# CAUCHY–SCHWARZ REGULARIZERS

**Sueda Taner**
ETH Zurich, Switzerland
taners@iis.ee.ethz.ch

**Ziyi Wang**
ETH Zurich, Switzerland
ziywang@student.ethz.ch

**Christoph Studer**
ETH Zurich, Switzerland
studer@ethz.ch

## ABSTRACT

We introduce a novel class of regularization functions, called Cauchy–Schwarz (CS) regularizers, which can be designed to induce a wide range of properties in solution vectors of optimization problems. To demonstrate the versatility of CS regularizers, we derive regularization functions that promote discrete-valued vectors, eigenvectors of a given matrix, and orthogonal matrices. The resulting CS regularizers are simple, differentiable, and can be free of spurious stationary points, making them suitable for gradient-based solvers and large-scale optimization problems. In addition, CS regularizers automatically adapt to the appropriate scale, which is, for example, beneficial when discretizing the weights of neural networks. To demonstrate the efficacy of CS regularizers, we provide results for solving underdetermined systems of linear equations and weight quantization in neural networks. Furthermore, we discuss specializations, variations, and generalizations, which lead to an even broader class of new and possibly more powerful regularizers.

## 1 INTRODUCTION

We focus on the design of novel regularization functions $\ell \colon \mathbb{R}^N \to \mathbb{R}_{\geq 0}$ that promote certain pre-defined properties on the solution vector(s) $\hat{\mathbf{x}} \in \mathbb{R}^N$ of regularized optimization problems

$$\hat{\mathbf{x}} \in \arg \min_{\mathbf{x} \in \mathbb{R}^N} f(\mathbf{x}) + \lambda \ell(\mathbf{x}), \tag{1}$$

where $f \colon \mathbb{R}^N \to \mathbb{R}$ is an objective function and $\lambda \in \mathbb{R}_{\geq 0}$ a regularization parameter. One instance of such an optimization problem arises in the binarization of neural-network weights, where the solution(s) of (1) are the network's weights that should be binary-valued $\hat{\mathbf{x}} \in \{-\alpha, +\alpha\}^N$, but with the appropriate scale $\alpha \in \mathbb{R}$ automatically chosen by the regularizer—the scale $\alpha$ can then be absorbed into the activation function.

### 1.1 CONTRIBUTIONS

We propose *Cauchy–Schwarz (CS) regularizers*, a novel class of regularization functions that can be designed to impose a wide range of properties. We derive concrete examples of CS regularization functions that promote discrete-valued vectors (e.g., binary- and ternary-valued vectors), eigenvectors of a given matrix, and matrices with orthogonal columns. The resulting regularizers are (i) simple, (ii) automatically determine the appropriate scale, (iii) often free of any spurious stationary points, and (iv) differentiable, which enables the use of (stochastic) gradient-based numerical solvers that make them suitable to be used in large-scale optimization problems. In addition, we discuss a variety of specializations, variations, and generalizations, which allow for the design of an even broader class of new and possibly more powerful regularization functions. Finally, we showcase the efficacy and versatility of CS regularizers for solving underdetermined systems of linear equations and neural network weight binarization and ternarization. All proofs and additional experimental results are relegated to the appendices in the supplementary material. The code for our numerical experiments is available under `https://github.com/IIP-Group/CS_regularizers`.

## 1.2 NOTATION

Column vectors and matrices are written in boldface lowercase and uppercase letters, respectively. The entries of a vector $\mathbf{x} \in \mathbb{R}^N$ are $[\mathbf{x}]_n = x_n$, $n = 1, \ldots, N$, and transposition is $\mathbf{x}^\mathrm{T}$. The $N$-dimensional all-zeros vector is $\mathbf{0}_N$, and the all-ones vector $\mathbf{1}_N$; we omit the dimension $N$ if it is clear from the context. The inner product between the vectors $\mathbf{x}, \mathbf{y} \in \mathbb{R}^N$ is $\langle \mathbf{x}, \mathbf{y} \rangle = \mathbf{x}^\mathrm{T} \mathbf{y}$, and linear dependence is denoted by

$$\mathbf{x} \sim \mathbf{y} \iff \exists (a_1, a_2) \in \mathbb{R}^2 \backslash \{(0,0)\} : a_1 \mathbf{x} = a_2 \mathbf{y}. \tag{2}$$

For $p \geq 1$, the $p$-norm of a vector $\mathbf{x} \in \mathbb{R}^N$ is $\|\mathbf{x}\|_p \triangleq (\sum_{n=1}^N |x_n|^p)^{1/p}$, and we will frequently use the shorthand notation $[\![\mathbf{x}]\!]^p \triangleq \sum_{n=1}^N x_n^p$ (note the absence of absolute values). The entry on the $n$th row and $k$th column of a matrix $\mathbf{X}$ is $X_{n,k}$, the Frobenius norm is $\|\mathbf{X}\|_\mathrm{F}$, and columnwise vectorization is $\mathrm{vec}(\mathbf{X})$. The $N \times N$ identity matrix is $\mathbf{I}_N$ and the $M \times N$ all-zero matrix is $\mathbf{0}_{M \times N}$.

## 1.3 RELEVANT PRIOR ART

Semidefinite relaxation (SDR) can be used for solving optimization problems with binary-valued solutions (Luo et al., 2010). Since SDR requires lifting (i.e., increasing the dimension of the original problem size), solving such problems quickly results in prohibitive complexity, even for moderate-sized problems. As a remedy, non-lifting-based SDR approximations were proposed in (Shah et al., 2016; Castañeda et al., 2017). These methods utilize biconvex relaxation that scales better to larger optimization problems. Convex non-lifting-based approaches were also proposed for recovering binary-valued solutions from linear measurements using $\ell^\infty$-norm regularization (Mangasarian & Recht, 2011). In contrast to such methods, the proposed CS regularizers are (i) differentiable, which enables their use together with differentiable objective functions and any (stochastic) gradient-based numerical solver, and (ii) can be specialized to impose a wider range of different structures.

Vector discretization is widely used for neural network parameter quantization (Hubara et al., 2018). Regularization-free approaches, e.g., the method from Rastegari et al. (2016), perform neural network binarization by simply quantizing the weights and adapting their scale to their average absolute value. Approaches that utilize projections onto discrete sets within gradient-descent-based methods have been proposed in Hou et al. (2016); Leng et al. (2018). In contrast, the proposed CS regularizers are differentiable and automatically adapt their scale to the appropriate magnitude of the solution vectors.

The prior art describes numerous vector discretization methods that rely on regularization functions. The methods in Hung et al. (2015); Tang et al. (2017); Wess et al. (2018); Bai et al. (2019); Darabi et al. (2019); Choi et al. (2020); Yang et al. (2021); Razani et al. (2021); Xu et al. (2023) use regularization functions related[1] to the form $\ell(\mathbf{x}, \beta) = \sum_{n=1}^N (x_n^2 - \beta)^2$ for $N$-dimensional vectors $\mathbf{x} \in \mathbb{R}^N$ and either fix the magnitude $\beta^2$ (e.g., $\beta = 1$) or learn this additional parameter during gradient descent. Another strain regularization functions that utilizes trigonometric functions related[2] to the form $\ell(\mathbf{x}, \beta) = \sum_{n=1}^N \sin^2(\beta \pi x_n)$ have been proposed in Naumov et al. (2018); Elthakeb et al. (2020); Solodskikh et al. (2022). In contrast to all of the above regularization functions, the proposed CS regularizers (i) do not introduce additional trainable parameters while still being able to automatically adapt their scale to the vectors' magnitude and (ii) can be designed to promote a wider range of structures. Furthermore, the proposed CS regularizers include the regularization functions of Tang et al. (2017); Darabi et al. (2019) as a special case; see App. B.3.

The recovery of matrices with orthogonal columns finds, for example, use in the orthogonal Procrustes problem: $\hat{\mathbf{X}} = \arg\min_{\mathbf{X} \in \mathbb{R}^{N \times K}} \|\mathbf{X}\mathbf{A} - \mathbf{B}\|_\mathrm{F}$ subject to $\mathbf{X}^\mathrm{T}\mathbf{X} = \mathbf{I}_K$ for given matrices $\mathbf{A}$ and $\mathbf{B}$. A closed-form solution to this problem is given by $\hat{\mathbf{X}} = \mathbf{U}\mathbf{V}^T$, where $\mathbf{U}$ and $\mathbf{V}$ are the left- and right-singular matrices, respectively, of the matrix $\mathbf{B}\mathbf{A}^\mathrm{T}$ (Schönemann, 1966). Notably, this formulation constrains the columns of $\mathbf{X}$ to be *orthonormal*. A more general problem that relaxes this constraint to *orthogonal* columns with arbitrary norms was introduced in Everson (1998) along

---

[1]Some methods, e.g., Darabi et al. (2019), use non-differentiable regularizers with $||x_n| - \beta|$ instead of $(x_n^2 - \beta)^2$, while others, e.g., Xu et al. (2023), use regularizers of the form $\gamma \|\mathbf{x} - \alpha \operatorname{sign}(\mathbf{x})\|$ and introduce additional scaling factors.

[2]The method in Naumov et al. (2018) fixes the scale, while Elthakeb et al. (2020) utilizes a trainable parameter; the regularizer in Solodskikh et al. (2022) introduces an additional differentiable regularizer that imposes finite range.

with iterative solution algorithms. In contrast to these approaches, CS regularizers can be used to solve Procrustes problems for matrices with orthogonal columns of arbitrary scale, *without* relying on matrix decomposition methods, such as the singular value, polar, or QR decomposition.

Promoting eigenvectors of a matrix is useful in applications where aligning a vector with these eigenvectors is desirable. For example, in principal component analysis (Jolliffe, 2002), data is projected onto the principal eigenvectors for dimensionality reduction. Encouraging this alignment can aid feature selection and thus improve dimensionality reduction. In this context, CS regularizers offer an advantage by promoting eigenvectors without explicitly computing eigenvalues or eigenvectors.

An instance of a CS regularizer was proposed in Ehrhardt & Arridge (2014), which is a regularization function based on the gradients of two vector-valued functions to measure how far these functions are from being parallel. In contrast, we present a general framework for designing a broad class of CS regularizers, encompassing the one from Ehrhardt & Arridge (2014) as a special case.

We finally note that the CS divergence (Principe, 2010) was used as a regularizer for improving variational autoencoders in Tran et al. (2022) and for promoting fairness in machine learning models in Liu et al. (2025). In contrast, we use the CS *inequality* (Steele, 2004) to design new regularization functions that can—among many other structures—be used to promote discrete-valued vectors (e.g., binary or ternary), eigenvectors to a given matrix, and matrices with orthogonal columns.

## 2 CAUCHY–SCHWARZ REGULARIZERS

In this section, we introduce the general recipe for deriving CS regularizers. We then use this recipe to design specific CS regularization functions that promote discrete-valued (e.g., binary and ternary) vectors, eigenvectors of a given matrix, and matrices with orthogonal columns.

### 2.1 THE RECIPE

The following result is an immediate consequence of the CS inequality (Steele, 2004) and provides a recipe for the design of a wide range of regularization functions; a short proof is provided in App. A.1.

**Proposition 1.** *Fix two vector-valued functions* $\mathbf{g}, \mathbf{h} : \mathbb{R}^N \to \mathbb{R}^M$ *and define the set*

$$\mathcal{X} \triangleq \left\{ \tilde{\mathbf{x}} \in \mathbb{R}^N : \mathbf{g}(\tilde{\mathbf{x}}) \sim \mathbf{h}(\tilde{\mathbf{x}}) \right\}. \tag{3}$$

*Then, the nonnegative regularization function*

$$\ell(\mathbf{x}) \triangleq \|\mathbf{g}(\mathbf{x})\|_2^2 \|\mathbf{h}(\mathbf{x})\|_2^2 - |\langle \mathbf{g}(\mathbf{x}), \mathbf{h}(\mathbf{x}) \rangle|^2 \tag{4}$$

*is zero if and only if (iff)* $\mathbf{x} \in \mathcal{X}$.

We call regularization functions derived from Proposition 1 *CS regularizers*. While CS regularizers are guaranteed to be (i) nonnegative and (ii) zero iff $\mathbf{x} \in \mathcal{X}$, it is also desirable for gradient-based numerical solvers that these regularizers do not exhibit any *spurious stationary points*.

**Definition 1.** *A spurious stationary point is a vector* $\mathbf{x} \notin \mathcal{X}$ *for which* $\nabla \ell(\mathbf{x}) = \mathbf{0}$.

Note that functions that do not have any spurious stationary points are also known as *invex* (Ben-Israel & Mond, 1986). In other words, all stationary points of invex functions are also global minima; this property can be useful to develop problem-specific algorithms and to analyze their convergence (see, e.g., Barik et al. (2023); Pinilla et al. (2022) and the references therein).

Whether or not a CS regularizer has spurious stationary points depends on the specific choice of the two functions $\mathbf{g}$ and $\mathbf{h}$. Nonetheless, even if a CS regularizer has spurious stationary points, it may still accomplish the desired goal. We conclude by noting that any vector $\mathbf{x}$ for which $\mathbf{g}(\mathbf{x}) = \mathbf{0}$ or $\mathbf{h}(\mathbf{x}) = \mathbf{0}$ will minimize (4).

The following result will be useful below when we analyze properties of specific CS regularizers; a short proof is given in App. A.2.

**Lemma 1.** *Fix two vector-valued functions* $\mathbf{g}, \mathbf{h} : \mathbb{R}^N \to \mathbb{R}^M$. *Then, the following equalities hold:*

$$\ell(\mathbf{x}) = \|\mathbf{g}(\mathbf{x})\|_2^2 \min_{\beta \in \mathbb{R}} \|\beta \mathbf{g}(\mathbf{x}) - \mathbf{h}(\mathbf{x})\|_2^2 = \|\mathbf{h}(\mathbf{x})\|_2^2 \min_{\beta \in \mathbb{R}} \|\mathbf{g}(\mathbf{x}) - \beta \mathbf{h}(\mathbf{x})\|_2^2. \tag{5}$$

Lemma 1 will be used to highlight the important *auto-scale property* CS regularizers, since setting $\beta$ to its optimal value in (5) leads exactly to the regularization function in (4), as the optimization problem in $\beta$ is continuous, quadratic, and has a closed-form solution.

## 2.2 Recovering Discrete-Valued Vectors

From Proposition 1, we can derive a range of *differentiable* CS regularizers that, when minimized, promote discrete-valued vectors. This can be accomplished by using entry-wise polynomials for the functions $\mathbf{g}$ and $\mathbf{h}$. We next show three concrete and practically useful examples.

### 2.2.1 Symmetric Binary

Define $\mathbf{g}(\mathbf{x}) \triangleq [x_1^2, \ldots, x_N^2]^{\mathrm{T}}$ and $\mathbf{h}(\mathbf{x}) \triangleq \mathbf{1}_N$. Then, Proposition 1 yields the following CS regularizer that promotes symmetric binary-valued vectors; see App. A.3 for the derivation.

**Regularizer 1** (Symmetric Binary). *Let $\mathbf{x} \in \mathbb{R}^N$ and define*

$$\ell_{\mathrm{bin}}(\mathbf{x}) \triangleq N[\![\mathbf{x}]\!]^4 - \left([\![\mathbf{x}]\!]^2\right)^2. \tag{6}$$

*Then, the nonnegative function in (6) is only zero for symmetric binary-valued vectors, i.e., iff $\mathbf{x} \in \{-\alpha, +\alpha\}^N$ for any $\alpha \in \mathbb{R}$. Furthermore, $\ell_{\mathrm{bin}}(\mathbf{x})$ does not have any spurious stationary points.*

To gain insight into Regularizer 1, we invoke Lemma 1 and obtain

$$\ell_{\mathrm{bin}}(\mathbf{x}) = N \min_{\beta \in \mathbb{R}_{\geq 0}} \sum_{n=1}^{N} \left((x_n - \sqrt{\beta})(x_n + \sqrt{\beta})\right)^2. \tag{7}$$

This equivalence implies that, for a given vector $\mathbf{x}$, Regularizer 1 is the right-hand-side total square error in (7), but with optimally chosen scale $\alpha \triangleq \sqrt{\beta}$; this is the *auto-scale property* of CS regularizers. In other words, the CS regularizer implicitly and automatically adapts its scale $\alpha$ to the scale of every argument $\mathbf{x}$. Furthermore, this CS regularizer is zero iff $\mathbf{x} \in \{-\alpha, \alpha\}^N$ for some $\alpha \in \mathbb{R}$, as only $x_n = +\alpha$ or $x_n = -\alpha$ for $n = 1, \ldots, N$ allows the right-hand-side of (7) to be zero.

We showcase the efficacy of $\ell_{\mathrm{bin}}$ for the recovery of binary-valued solutions in Section 3.1 and compare its advantages to existing binarizing regularizers (cf. Section 1.3), such as being differentiable, scale-adaptive, and free of additional optimization parameters, in App. C.1.

### 2.2.2 One-Sided Binary

Define $\mathbf{g}(\mathbf{x}) \triangleq [x_1^2, \ldots, x_N^2]^{\mathrm{T}}$ and $\mathbf{h}(\mathbf{x}) \triangleq [x_1, \ldots, x_N]^{\mathrm{T}}$. Then, Proposition 1 yields the following CS regularizer that promotes one-sided binary-valued vectors; see App. A.4 for the derivation.

**Regularizer 2** (One-Sided Binary). *Let $\mathbf{x} \in \mathbb{R}^N$ and define*

$$\ell_{\mathrm{osb}}(\mathbf{x}) \triangleq [\![\mathbf{x}]\!]^2 [\![\mathbf{x}]\!]^4 - \left([\![\mathbf{x}]\!]^3\right)^2. \tag{8}$$

*Then, the nonnegative function in (8) is only zero for one-sided binary-valued vectors, i.e., iff $\mathbf{x} \in \{0, \alpha\}^N$ for any $\alpha \in \mathbb{R}$. Furthermore, $\ell_{\mathrm{osb}}(\mathbf{x})$ does not have any spurious stationary points.*

To gain insight into Regularizer 2, we invoke Lemma 1 and obtain

$$\ell_{\mathrm{osb}}(\mathbf{x}) = [\![\mathbf{x}]\!]^2 \min_{\beta \in \mathbb{R}} \sum_{n=1}^{N} \left(x_n(x_n - \beta)\right)^2. \tag{9}$$

Once again, we observe this CS regularizer's auto-scale property and see that only vectors of the form $\mathbf{x} \in \{0, \alpha\}^N$ for some $\alpha \in \mathbb{R}$ minimize (9).

### 2.2.3 Symmetric Ternary

Define $\mathbf{g}(\mathbf{x}) \triangleq [x_1^3, \ldots, x_N^3]^{\mathrm{T}}$ and $\mathbf{h}(\mathbf{x}) \triangleq [x_1, \ldots, x_N]^{\mathrm{T}}$. Then, Proposition 1 yields the following CS regularizer that promotes symmetric ternary-valued vectors; see App. A.5 for the derivation.

**Regularizer 3** (Symmetric Ternary). *Let $\mathbf{x} \in \mathbb{R}^N$ and define*

$$\ell_{\mathrm{ter}}(\mathbf{x}) \triangleq [\![\mathbf{x}]\!]^2 [\![\mathbf{x}]\!]^6 - \left([\![\mathbf{x}]\!]^4\right)^2. \tag{10}$$

*Then, the nonnegative function in (10) is only zero for symmetric ternary-valued vectors, i.e., iff $\mathbf{x} \in \{-\alpha, 0, +\alpha\}^N$ for any $\alpha \in \mathbb{R}$. Furthermore, $\ell_{\mathrm{ter}}(\mathbf{x})$ does not have any spurious stationary points.*

To gain insight into Regularizer 3, we invoke Lemma 1 and obtain

$$\ell_{\text{ter}}(\mathbf{x}) = [\![\mathbf{x}]\!]^2 \min_{\beta \in \mathbb{R}_{\geq 0}} \sum_{n=1}^{N} \left( x_n (x_n - \sqrt{\beta})(x_n + \sqrt{\beta}) \right)^2. \tag{11}$$

As above, we observe this CS regularizer's auto-scale property and see that only vectors of the form $\mathbf{x} \in \{-\alpha, 0, +\alpha\}^N$ for some $\alpha \in \mathbb{R}$ minimize (11).

The CS regularizers introduced so far promote binary- or ternary-valued vectors; a visualization of their loss landscapes in two dimensions is provided in App. F. In App. B.1, we detail an approach that generalizes CS regularizers to a symmetric, discrete-valued set with $2^B$ equispaced entries. In addition, all of the CS regularizers introduced above involve polynomials of higher (e.g., quartic) order, leading to potential numerical stability issues. In App. B.2, we propose alternative symmetric binarization regularizers that avoid such issues; similar alternative regularization functions can be derived for the other discretization regularizers.

## 2.3 RECOVERING EIGENVECTORS OF A GIVEN MATRIX

All CS regularizers introduced so far promote vectors with discrete-valued entries. In order to demonstrate the versatility of Proposition 1, we now propose a CS regularizer that promotes vectors that are eigenvectors of a given (and fixed) matrix $\mathbf{C} \in \mathbb{R}^{N \times N}$.

Define $\mathbf{g}(\mathbf{x}) \triangleq \mathbf{C}\mathbf{x}$ and $\mathbf{h}(\mathbf{x}) = \mathbf{x}$. Then, Proposition 1 yields the following CS regularizer that promotes eigenvectors of the matrix $\mathbf{C}$; see App. A.6 for the derivation.

**Regularizer 4** (Eigenvector). *Fix* $\mathbf{C} \in \mathbb{R}^{N \times N}$, *let* $\mathbf{x} \in \mathbb{R}^N$, *and define*

$$\ell_{\text{eig}}(\mathbf{x}) \triangleq \|\mathbf{C}\mathbf{x}\|_2^2 \|\mathbf{x}\|_2^2 - (\mathbf{x}^T \mathbf{C}\mathbf{x})^2 \tag{12}$$

*Then, the nonnegative function in (12) is only zero for eigenvectors of* $\mathbf{C}$ *and the all-zeros vector.*

To gain insight into Regularizer 4, we invoke Lemma 1 and obtain

$$\ell_{\text{eig}}(\mathbf{x}) = [\![\mathbf{x}]\!]^2 \min_{\beta \in \mathbb{R}} \|\mathbf{C}\mathbf{x} - \beta\mathbf{x}\|_2^2. \tag{13}$$

As above, we observe this CS regularizer's auto-scale property and see that only scaled eigenvectors of the matrix $\mathbf{C}$ minimize (13).

## 2.4 RECOVERING MATRICES WITH ORTHOGONAL COLUMNS

Finally, we demonstrate that Proposition 1 can also be used to promote structure in matrices. The following CS regularizer promotes matrices $\mathbf{X} \in \mathbb{R}^{N \times K}$ with $K \leq N$ to have orthogonal columns.

Define $\mathbf{g}(\mathbf{X}) \triangleq \text{vec}(\mathbf{X}^\mathsf{T}\mathbf{X})$ and $\mathbf{h}(\mathbf{x}) \triangleq \text{vec}(\mathbf{I}_K)$. Then, Proposition 1 yields the following CS regularizer that promotes matrices with orthogonal columns; see App. A.7 for the derivation.

**Regularizer 5** (Matrix with Orthogonal Columns). *Let* $\mathbf{X} \in \mathbb{R}^{N \times K}$ *with* $K \leq N$ *and define*

$$\ell_{\text{om}}(\mathbf{X}) \triangleq K \|\mathbf{X}^\mathsf{T}\mathbf{X}\|_{\text{F}}^2 - \|\mathbf{X}\|_{\text{F}}^4. \tag{14}$$

*Let* $\mathbf{X} = \mathbf{U}\mathbf{S}\mathbf{V}^\mathsf{T}$ *be the singular value decomposition of* $\mathbf{X}$. *Then, we equivalently have,*

$$\ell_{\text{om}}(\mathbf{X}) \triangleq K \left( \sum_{k=1}^K S_{k,k}^4 \right) - \left( \sum_{k=1}^K S_{k,k}^2 \right)^2, \tag{15}$$

*which is the symmetric binarizer from (6) applied to the singular values of* $\mathbf{X}$. *The nonnegative function in (14) is only zero for matrices* $\mathbf{X}$ *with pairwise orthogonal columns of equal length, i.e., iff* $\mathbf{X}^\mathsf{T}\mathbf{X} = \alpha\mathbf{I}_K$ *for* $\alpha > 0$. *Furthermore,* $\ell_{\text{om}}(\mathbf{X})$ *does not have any spurious stationary points.*

To gain insight into Regularizer 5, we invoke Lemma 1 and obtain

$$\ell_{\text{om}}(\mathbf{X}) = K \min_{\beta \in \mathbb{R}} \|\text{vec}(\mathbf{X}^\mathsf{T}\mathbf{X}) - \beta\text{vec}(\mathbf{I}_K)\|_2^2. \tag{16}$$

Once again, we observe this CS regularizer's auto-scale property and see that only matrices $\mathbf{X}$ with orthogonal columns of the same norm minimize (16). Note that we have developed a regularizer that promotes the same norm for all columns of $\mathbf{X}$ for simplicity. However, one could modify $\mathbf{h}(\mathbf{x})$ to promote, for example, a certain user-defined ratio between the norms of the columns.

## 2.5 GENERALIZATIONS AND VARIATIONS

Proposition 1 and the underlying ideas enable the design of a much broader range of regularizers. We now propose one possible generalization, where we replace the CS inequality utilized in Proposition 1 with Hölder's inequality (Hölder, 1889), resulting in the following recipe for *Hölder regularizers*.

**Proposition 2.** *Fix two vector-valued functions* $\mathbf{g}, \mathbf{h}\colon \mathbb{R}^N \to \mathbb{R}^N$ *and define* $\mathcal{X}$ *as in (3). Let* $p, q \geq 1$ *so that* $\frac{1}{p} + \frac{1}{q} = 1$ *and let* $r > 0$. *Then, the nonnegative function*

$$\breve{\ell}(\mathbf{x}) \triangleq \|\mathbf{g}(\mathbf{x})\|_p^r \|\mathbf{h}(\mathbf{x})\|_q^r - |\langle \mathbf{g}(\mathbf{x}), \mathbf{h}(\mathbf{x})\rangle|^r \tag{17}$$

*is zero iff* $\mathbf{x} \in \mathcal{X}$.

Note that Proposition 1 is a special case of Proposition 2 by setting $p = q = r = 2$.

We include more generalizations and variations in App. B. Specifically, we propose a CS regularizer promoting vectors with symmetric equispaced discrete values in App. B.1, bounded CS regularizers for vector binarization in App. B.2, CS regularizers for discrete-valued vectors with fixed scale in App. B.3, non-differentiable variants of CS regularizers in App. B.4, and a generalization of CS regularizers that is invariant to the scale of the functions $\mathbf{g}$ and $\mathbf{h}$ in App. B.5. We conclude by noting that many other CS or Hölder regularizers can be derived when combining the above ideas and those put forward in App. B. We also note that most of these results can be generalized to complex-valued vectors. A detailed investigation of such generalizations and variations is left for future work.

## 3 APPLICATION EXAMPLES

We now show several application examples for CS regularizers for vector discretization, recovery of eigenvectors of a given matrix, recovering matrices with orthogonal columns, and quantization of neural network weights.

### 3.1 RECOVERING DISCRETE-VALUED VECTORS

The proposed CS regularizers enable the recovery of binary- and ternary-valued vectors from underdetermined linear systems of equations. To this end, we solve systems of linear equations of the form $\mathbf{b} = \mathbf{A}\mathbf{x}$, where $\mathbf{A} \in \mathbb{R}^{M \times N}$ has i.i.d. standard normal entries and $M < N$. We create vectors $\mathbf{x}^\star \in \mathbb{R}^N$, whose entries are chosen i.i.d. with uniform probability from $\{-1, +1\}$ for symmetric binary and from $\{0, +1\}$ for one-sided binary, and with probability $0.25, 0.5, 0.25$ from $\{-1, 0, +1\}$, respectively, for ternary-valued vectors. Then, we calculate $\mathbf{b} = \mathbf{A}\mathbf{x}^\star$, and we try to recover the vector $\mathbf{x}^\star$ from $\mathbf{b}$ by solving optimization problems of the form

$$\hat{\mathbf{x}} \in \arg\min_{\tilde{\mathbf{x}} \in \mathbb{R}^N} \ell(\tilde{\mathbf{x}}) \quad \text{subject to } \mathbf{b} = \mathbf{A}\tilde{\mathbf{x}} \tag{18}$$

using a projected gradient descent algorithm—specifically, FISTA with backtracking (Beck & Teboulle, 2009; Goldstein)[3]. Here, $\ell(\tilde{\mathbf{x}})$ are the CS regularizers from Section 2.2. We fix $N = 100$ and vary $M$ between 30 and 90.[4] We declare success for recovering $\mathbf{x}^\star$ if the returned solution $\hat{\mathbf{x}}$ satisfies $\|\hat{\mathbf{x}} - \mathbf{x}^\star\|_2 / \|\mathbf{x}^\star\|_2 \leq 10^{-2}$. Fig. 1 shows the success probabilities with respect to the undersampling ratio $\gamma$ along with error bars calculated from the standard error of the mean.

**Symmetric Binary** We first recover symmetric binary-valued solutions using Regularizer 1 with $\ell_{\text{bin}}$ from (6). For this scenario, Mangasarian & Recht (2011) showed that $\ell^\infty$-norm minimization recovers the binary-valued solution as long as $\gamma = M/N$ satisfies $\gamma > 0.5$ and $N$ approaches infinity. Thus, our baseline is $\ell^\infty$-norm minimization, which we solve with Douglas–Rachford splitting (Eckstein & Bertsekas, 1992) as in Studer et al. (2015). Fig. 1a shows the success rate for $\ell_{\text{bin}}$ and $\ell^\infty$-norm minimization with respect to the undersampling ratio $\gamma$. For $\ell_{\text{bin}}$ minimization, we observed that different initializations have an impact on the success rate since the objective is non-convex. Thus, we allow at most 10 random initializations of projected gradient descent. We note

---

[3]We run projected gradient descent and the baseline algorithms for a maximum of $10^4$ iterations.

[4]We also study the impact of the sparsity of $\mathbf{x}^\star$ while the number of measurements $M$ is fixed in App. C.4. For each $M$, we randomly generate 1000 problem instances and report the average success probability.

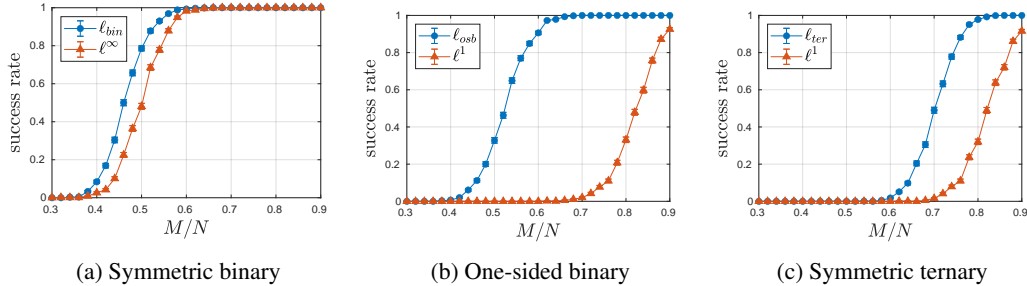

(a) Symmetric binary       (b) One-sided binary       (c) Symmetric ternary

Figure 1: Probability of success for recovering vectors with (a) binary, (b) one-sided binary, and (c) symmetric ternary values dependent on the undersampling ratio $M/N$.

that multiple initializations of $\ell^\infty$-norm minimization do not affect its success rate as the problem is convex. We see from Fig. 1a that $\ell_{\text{bin}}$ minimization has, for any undersampling ratio $\gamma$, a higher probability of successfully recovering the true solution than $\ell^\infty$-norm minimization.

We discuss the advantages of $\ell_{\text{bin}}$ over existing binarizing regularizers in App. C.1. We also provide a comparison of the success rates of $\ell_{\text{bin}}$ and $\ell^\infty$-norm minimization with existing binarizing regularizers in App. C.2, and observe that $\ell_{\text{bin}}$ achieves the highest success rate. Moreover, in App. C.3, we provide the success rates of other binarizing CS regularizer variants (e.g., the Hölder, non-differentiable, and scale-invariant regularizers), where $\ell_{\text{bin}}$, once again, achieves best performance.

In App. C.7, we showcase yet another application for the $\ell_{\text{bin}}$ regularizer. Specifically, we use it to find approximate solutions to the *weighted maximum cut* (MAX-CUT) problem (Commander, 2009). Our simulation results in App. C.7 show that our approach does not identify optimal solutions for large graphs, but significantly improves the objective values compared to the initialization.

**One-Sided Binary** We now recover vectors with one-sided binary-valued entries using Regularizer 2 with $\ell_{\text{osb}}$ from (8). In this experiment, we ran projected gradient descent for only one random initialization. Our baseline for this scenario is $\ell^1$-norm minimization (Cai et al., 2009), as the generated vectors are sparse with half of the entries being nonzero (on average). We solve the $\ell^1$-norm minimization problem with Douglas–Rachford splitting. Fig. 1b demonstrates that $\ell_{\text{osb}}$ minimization significantly outperforms $\ell^1$-norm minimization for all undersampling ratios.

**Symmetric Ternary** We also recover vectors with symmetric ternary-valued entires using Regularizer 3 with $\ell_{\text{ter}}$ from (10). In this experiment, we ran projected gradient descent for only one random initialization. We use $\ell^1$-norm minimization as our baseline as the generated vectors are sparse with half of the entries being nonzero (on average). Fig. 1c demonstrates that $\ell_{\text{ter}}$ minimization significantly outperforms $\ell^1$-norm minimization for all undersampling ratios.

We provide similar simulation results for symmetric two-bit-valued solution recovery in App. C.5.

## 3.2 RECOVERING EIGENVECTORS OF A MATRIX

The proposed CS regularizers also enable the recovery of eigenvectors of a given (and fixed) matrix. We once again consider a system of linear equations of the form $\mathbf{b} = \mathbf{A}\mathbf{x}$, where $\mathbf{A} \in \mathbb{R}^{M \times N}$ has i.i.d. standard normal entries and $M < N$. We create vectors $\mathbf{x}^\star \in \mathbb{R}^N$ by uniform randomly choosing an eigenvector of a matrix $\mathbf{C} \in \mathbb{R}^{N \times N}$ with i.i.d. standard normal entries. We calculate $\mathbf{b} = \mathbf{A}\mathbf{x}^\star$ and solve (18) using Regularizer 4 with $\ell_{\text{eig}}$. As a baseline, we consider minimizing $\ell_\mu(\tilde{\mathbf{x}}, \tilde{\mu}) \triangleq \|\mathbf{C}\tilde{\mathbf{x}} - \tilde{\mu}\tilde{\mathbf{x}}\|_2^2$ subject to $\mathbf{b} = \mathbf{A}\tilde{\mathbf{x}}$, similarly to (18), with an additional optimization parameter (i.e., $\tilde{\mu}$) compared to minimizing $\ell_{\text{eig}}(\tilde{\mathbf{x}})$. We use FISTA as in Section 3.1 for both algorithms. Fig. 7 in App. C.6 demonstrates that the success rate of $\ell_{\text{eig}}$ remains to be consistently above 0.8, approaching 1 at an $M/N$-ratio of 0.5, and surpassing that of the baseline that minimizes $\ell_\mu$.

## 3.3 RECOVERING MATRICES WITH ORTHOGONAL COLUMNS

We now demonstrate that the proposed regularizers can also impose structure to matrices. To this end, we consider a system of linear equations $\mathbf{AX} = \mathbf{B}$, where $\mathbf{A} \in \mathbb{R}^{M \times N}$ has i.i.d. standard normal entries with $M < N$, and $\mathbf{X} \in \mathbb{R}^{N \times K}$ with $\mathbf{X}^{\mathsf{T}}\mathbf{X} = \mathbf{I}_K$. We solve

$$\hat{\mathbf{X}} \in \arg \min_{\tilde{\mathbf{X}} \in \mathbb{R}^{N \times K}} \ell_{\text{om}}(\tilde{\mathbf{X}}) \quad \text{subject to } \mathbf{A}\tilde{\mathbf{X}} = \mathbf{B} \tag{19}$$

using FISTA as in Section 3.1 with a maximum number of 1000 iterations. We have observed that for $N = K = 10$ and $N = K = 100$ and for values of $M$ such that $M/N \in [0.1, 1]$, the output of FISTA was *always* an orthogonal matrix, i.e., the success rate is always one.

## 3.4 QUANTIZING NEURAL NETWORK WEIGHTS

We now provide another application example for binarizing and ternarizing neural network weights. Our goal is to further highlight the simplicity, versatility, and effectiveness of CS regularizers.

### 3.4.1 METHOD

Our weight quantization procedure consists of three steps: (i) training with CS regularizers, (ii) weight quantization, and (iii) continued training of remaining parameters. We detail these steps below. In order to demonstrate solely the impact of CS regularizers, we neither modify the neural network architecture (e.g., we do not alter the layers or activations) nor the forward-backward propagation stages, since we do not introduce any non-differentiable operations during training.

**Step 1: Regularized Training** Let $\boldsymbol{\theta}$ denote the set of all parameters of a neural network and $L(\boldsymbol{\theta})$ the loss function for learning the network's task. We solve the following optimization problem:

$$\hat{\boldsymbol{\theta}} = \arg \min_{\boldsymbol{\theta}} L(\boldsymbol{\theta}) + \lambda \sum_{k=1}^{K} \eta_k \, \ell(\boldsymbol{\theta}_k). \tag{20}$$

Here, $\lambda \in \mathbb{R}_{\geq 0}$ is a regularization parameter, $\boldsymbol{\theta}_k$ is a vector consisting of the network weights which should share a common scaling factor (this can, for example, be an entire layer, one convolution kernel, or any other subset of network parameters), $\eta_k$ is the associated normalization factor (e.g., the reciprocal value of the dimension of $\boldsymbol{\theta}_k$), and $\ell$ can be any CS regularizer (e.g., $\ell_{\text{bin}}$ or $\ell_{\text{ter}}$).

After training the network parameters with the CS regularizers for a given number of epochs, the weights contained in the vectors $\boldsymbol{\theta}_k$ will be *concentrated* around $\{-\alpha_k, \alpha_k\}$ for symmetric binary-valued regularization or $\{-\alpha_k, 0, \alpha_k\}$ for symmetric ternary-valued regularization for some $\alpha_k > 0$.

**Step 2: Quantization** The goal is to quantize the regularized weights $\{\boldsymbol{\theta}_k\}_{k=1}^{K}$ from the previous step. In what follows, we describe the quantization procedure for one weight vector $\mathbf{w} = \boldsymbol{\theta}_k$.

We binarize the regularized weight vector $\mathbf{w}$ according to $\hat{\mathbf{w}} \triangleq \arg \min_{\mathbf{x} \in \mathcal{X}_{\text{bin}}} \|\mathbf{w} - \mathbf{x}\|_2^2$ with $\mathcal{X}_{\text{bin}} \triangleq \{\tilde{\mathbf{x}} \in \{-\alpha, \alpha\}^N : \alpha \in \mathbb{R}\}$ as in Rastegari et al. (2016), which is given by

$$\hat{w}_n = \alpha^\star \text{sign}(w_n), \, n = 1, \ldots, N, \text{ with } \alpha^\star = \|\mathbf{w}\|_1 / N. \tag{21}$$

We ternarize the regularized weight vector $\mathbf{w} \neq \mathbf{0}_N$ according to $\hat{\mathbf{w}} \triangleq \arg \min_{\mathbf{x} \in \mathcal{X}_{\text{ter}}} \|\mathbf{w} - \mathbf{x}\|_2^2$ with $\mathcal{X}_{\text{ter}} \triangleq \{\tilde{\mathbf{x}} \in \{-\alpha, 0, \alpha\}^N : \alpha \in \mathbb{R}\}$ as in Li et al. (2016), which is accomplished as follows: Let $\mathcal{I}_\tau = \{i : |w_i| \geq \tau\}$. Then, find the threshold that determines which entries of $\hat{\mathbf{w}}$ are nonzero as

$$\tau^\star = \arg \max_{\tau \in \{|w_i| : i \in \mathcal{I}\}} \frac{1}{|\mathcal{I}_\tau|} (\sum_{i \in \mathcal{I}_\tau} |w_i|)^2. \tag{22}$$

Finally, compute the ternarized vector as

$$\hat{w}_n = \begin{cases} \alpha^\star \text{sign}(w_n), & |w_n| \geq \tau^\star \\ 0, & \text{otherwise,} \end{cases} n = 1, \ldots, N, \text{ with } \alpha^\star = \frac{1}{|\mathcal{I}_\tau|} \sum_{i \in \mathcal{I}_\tau} |w_i|. \tag{23}$$

**Step 3: Training with Quantized Weights** After weight quantization, the number of trainable parameters is significantly reduced since we now have only one scale factor for a vector of quantized weights. Hence, we fix the signs of the weights and continue training only their shared scale factors alongside other tunable network parameters (e.g., biases, batch normalization parameters, etc.) without the use of CS regularizers and for a small number of epochs.

### 3.4.2 EXPERIMENTAL RESULTS

We conduct experiments on the benchmark datasets ImageNet (ILSVRC12) (Deng et al., 2009) and CIFAR-10 (Krizhevsky, 2009) for image classification using PyTorch (Paszke et al., 2019). We follow classical data augmentation strategies as detailed in App. D.1.

**Implementation** As in Rastegari et al. (2016); Qin et al. (2020); He et al. (2020), we regularize and quantize all network layers except for the first convolutional layer and the last fully-connected layer. For convolutional layers, we apply the CS regularizer to vectors consisting of the weights in all kernels that produce one output channel; this leads to one scaling factor for each output channel following the approach from Rastegari et al. (2016). For fully-connected layers, we use one CS regularizer for each row of the weight matrix; this leads to one scaling factor for each output feature. We set the weights $\eta_k$ in (20) to the reciprocal of the dimension of the corresponding weight vector $\boldsymbol{\theta}_k$.

For ImageNet, we use ResNet-18 (He et al., 2016), initialize the weights with a pretrained full-precision model from PyTorch, and train the network for 40 and 20 epochs in Steps 1 and 3, respectively, with a batch size of 1024. For CIFAR-10, we use ResNet-20, initialize the weights with a pretrained full-precision model from Idelbayev (2021) similarly to Qin et al. (2020), and train the network for 400 and 20 epochs in Steps 1 and 3, respectively, with a batch size of 128. For both datasets, we set $\lambda = 10$ for binarization and $\lambda = 10^5$ for ternarization. [5] We use the Adam optimizer (Kingma & Ba, 2017) with its learning rate initialized by $0.001$ and the cosine annealing learning rate scheduler (Loshchilov & Hutter, 2016).

**Weight Distribution** Fig. 2 illustrates the impact of CS regularizers on the network weights with histograms for one output channel of one convolutional layer based on training ResNet-18 on ImageNet. We observe in Fig. 2(a) that the weight distribution of the pretrained network resembles that of a zero-mean Gaussian distribution. Figures 2(b) and (c) reveal, as expected, that the weights are becoming more concentrated around binary values after 5 and 20 epochs of training with the regularizer $\ell_{\text{bin}}$, respectively. Figures 2(d) and (e) reveal a similar behavior when using $\ell_{\text{ter}}$.

**Performance Evaluation** In App. D.3, Tables 8-10, we provide the top-1 accuracy for our methods along with the full-precision baseline and various SOTA baselines that binarize or ternarize the weights of the network while the activations are left in full precision. All SOTA top-1 accuracy results in Tables 8-10 are taken from the corresponding papers.

For ImageNet, we observe from Tables 8 and Table 9 that for binary-valued weights, our approach outperforms SQ-BWN (Dong et al., 2017), BWN (Rastegari et al., 2016), HWGQ (Cai et al., 2017), PCCN (Gu et al., 2019), and the ternary TWN (Li et al., 2016), while the accuracy we achieve is $4.9\%$ lower than the best SOTA method ProxyBNN (He et al., 2020). For ternary-valued weights, our approach outperforms TWN and SQ-TWN (Dong et al., 2017), while the accuracy we achieve is $2.8\%$ lower than the SOTA method QIL (Jung et al., 2019).

For CIFAR-10, we observe from Table 10 that for binary-valued weights, our approach outperforms DoReFa-Net by a small margin and achieve the same performance as LQ-Net (Zhang et al., 2018), while the accuracy of our approach is $0.9\%$ lower than the SOTA methods DAQ (Kim et al., 2021) and LCR-BNN (Shang et al., 2022); these two methods are also the only methods outperforming our ternary-valued approach by $0.2\%$.

While some of the SOTA methods achieve better accuracy than our approach, our results (i) require a simpler training procedure[6] and (ii) showcase the potential of CS regularizers: We only have one step of regularized training with full-precision weights, a quantization step, and a second step of training with fewer parameters; this procedure does not require any additional storage at any stage of training. In contrast, all of the baseline methods retain both the quantized and full-precision values for the weights during training, and use the quantized weights in forward and backward propagation while the full-precision values are updated with the gradients calculated with respect to the quantized values. This results in additional storage. Moreover, to reduce the quantization error or to alleviate

---

[5]We chose the regularization parameter $\lambda$ empirically based on using $1/10$th of the training sets for validation. We have observed that changes by a factor of 10 in $\lambda$ do not have a substantial impact on the resulting accuracies. Please see Tables 3-6 in App. D.2 for an ablation study for varying $\lambda$.

[6]Please see Table 7 for the advantages/disadvantages of our training strategy compared to the SOTA methods.

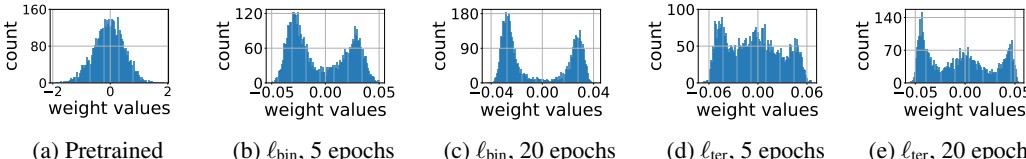

| (a) Pretrained | (b) $\ell_{\text{bin}}$, 5 epochs | (c) $\ell_{\text{bin}}$, 20 epochs | (d) $\ell_{\text{ter}}$, 5 epochs | (e) $\ell_{\text{ter}}$, 20 epochs |

Figure 2: Neural network weight histograms of one output channel of a convolutional layer in the pretrained ResNet-18 model and the model after regularized training with $\ell_{\text{bin}}$ or $\ell_{\text{ter}}$ over ImageNet.

the mismatch between forward and backward propagation, references Gu et al. (2019); Hu et al. (2018); Kim et al. (2021); He et al. (2020); Yang et al. (2019); Zhang et al. (2018); Jung et al. (2019) introduce more trainable parameters and Shang et al. (2022) proposes a regularization method that requires the construction of matrices that scale with the square of the number of features in one layer. Please see Table 7 in App. D.3 for a comparison of the additional variables introduced by each SOTA method.

We conclude by noting that the proposed CS regularizers could be combined with any of these existing approaches for possible further accuracy improvements.

## 4 LIMITATIONS

While the proposed CS regularizers provide a recipe for designing regularization functions with a wide variety of properties, they suffer from a range of limitations that we summarize next.

First and foremost, CS regularizers are typically nonconvex. While we have been able to prove that some of the proposed nonconvex CS regularizers are free of any spurious stationary points (and thus are *invex*), convergence to a global minimum depends on the combination of the objective, regularizer, and optimization algorithm. Thus, multiple random restarts of the optimizer might be necessary. Depending on the specific regularizer, a solution algorithm that exploits invexity, such as the one presented in Barik et al. (2023), could be utilized instead.

Furthermore, some of the CS regularizers involve higher-order polynomials (e.g., the ternarization regularizer in (10) involves eighth-order polynomials), which can result in poor convergence behavior, especially around their minimum. To counteract this issue, one can either resort to scale-invariant versions as outlined in App. B.5 or to non-differentiable variants as outlined in App. B.4. In addition, utilizing adaptive step-size selection methods and schedules that adapt (e.g., increase) the regularization parameter $\lambda$ over iterations could also be used to improve convergence.

Finally, we have only scratched the surface of many of the generalizations and variations put forward in Section 2.5 and in App. B. Besides that, we have investigated the efficacy of CS regularizers with only four example tasks, i.e., solving underdetermined systems of linear equations for recovering discrete-valued solutions, eigenvectors, and matrices with orthogonal columns, and neural network weight quantization with a simple training recipe, in Section 3. A thorough theoretical analysis and simulative study of alternative CS regularizers in a broader range of applications, as well as combining CS regularizers with sophisticated SOTA quantized neural network training procedures, such as the ones in He et al. (2020); Jung et al. (2019), are left for future work.

## 5 CONCLUSIONS

We have proposed Cauchy–Schwarz (CS) regularizers, a novel class of regularization functions that can be designed to promote a wide range of properties. We have derived example regularization functions that promote discrete-valued vectors, eigenvectors to matrices, or matrices with orthogonal columns, and we have outlined a range of specializations, variations, and generalizations that lead to an even broader class of new and possibly more powerful regularizers. For solving underdetermined systems of linear equations, we have shown that CS regularizers can outperform well-established baseline methods, such as $\ell^\infty$-norm or $\ell^1$-norm minimization. For weight quantization of neural networks, we have shown that utilizing CS regularizers enables one to achieve competitive accuracy to existing quantization methods while using a simple training procedure.

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

# Appendix: Cauchy–Schwarz Regularizers

## A  PROOFS AND DERIVATIONS

### A.1  PROOF OF PROPOSITION 1

From the Cauchy–Schwarz inequality (Steele, 2004) follows that

$$|\langle \mathbf{g}(\mathbf{x}), \mathbf{h}(\mathbf{x}) \rangle| \leq \|\mathbf{g}(\mathbf{x})\|_2 \|\mathbf{h}(\mathbf{x})\|_2. \tag{24}$$

Squaring both sides of (24) and rearranging terms leads to

$$0 \leq \|\mathbf{g}(\mathbf{x})\|_2^2 \|\mathbf{h}(\mathbf{x})\|_2^2 - |\langle \mathbf{g}(\mathbf{x}), \mathbf{h}(\mathbf{x}) \rangle|^2 \triangleq \ell(\mathbf{x}). \tag{25}$$

Equality in (24) holds iff $\mathbf{g}(\mathbf{x}) \sim \mathbf{h}(\mathbf{x})$, for which $\ell(\mathbf{x}) = 0$.

### A.2  PROOF OF LEMMA 1

Assume that $\mathbf{h}(\mathbf{x}) \neq \mathbf{0}$. Then,

$$\frac{\partial \|\mathbf{g}(\mathbf{x}) - \beta \mathbf{h}(\mathbf{x})\|_2^2}{\partial \beta} = 0 \implies \beta = \frac{\langle \mathbf{g}(\mathbf{x}), \mathbf{h}(\mathbf{x}) \rangle}{\|\mathbf{h}(\mathbf{x})\|_2^2}. \tag{26}$$

Plugging the right-hand-side into $\|\mathbf{g}(\mathbf{x}) - \beta \mathbf{h}(\mathbf{x})\|_2^2$ yields

$$\min_{\beta \in \mathbb{R}} \|\mathbf{g}(\mathbf{x}) - \beta \mathbf{h}(\mathbf{x})\|_2^2 = \|\mathbf{g}(\mathbf{x})\|_2^2 - \frac{|\langle \mathbf{g}(\mathbf{x}), \mathbf{h}(\mathbf{x}) \rangle|^2}{\|\mathbf{h}(\mathbf{x})\|_2^2}. \tag{27}$$

Multiplying both sides by $\|\mathbf{h}(\mathbf{x})\|_2^2$ results in

$$\ell(\mathbf{x}) = \|\mathbf{h}(\mathbf{x})\|_2^2 \min_{\beta \in \mathbb{R}} \|\mathbf{g}(\mathbf{x}) - \beta \mathbf{h}(\mathbf{x})\|_2^2. \tag{28}$$

If $\mathbf{h}(\mathbf{x}) = \mathbf{0}$, then (28) still holds. By swapping $\mathbf{g}(\mathbf{x})$ with $\mathbf{h}(\mathbf{x})$, the equalities in (5) follow.

### A.3  DERIVATION OF REGULARIZER 1

Regularizer 1 is minimized by vectors $\mathbf{x} \in \mathbb{R}^N$ that satisfy the linear dependence condition $\mathbf{g}(\mathbf{x}) \sim \mathbf{h}(\mathbf{x})$ for the specific choices $\mathbf{g}(\mathbf{x}) = [x_1^2, \ldots, x_N^2]^\mathsf{T}$ and $\mathbf{h}(\mathbf{x}) = \mathbf{1}_N$. We have

$$\mathbf{g}(\mathbf{x}) \sim \mathbf{h}(\mathbf{x}) \iff \exists (a_1, a_2) \in \mathbb{R}^2 \backslash \{(0,0)\} : a_1 x_n^2 = a_2, \quad n = 1, \ldots, N. \tag{29}$$

If $a_1 \neq 0$, then $x_n^2 = a_2/a_1$ which implies $x_n = \pm \alpha$, $n = 1, \ldots, N$, for some $\alpha \in \mathbb{R}$. If $a_1 = 0$ then $a_2 \neq 0$, so the condition $a_1 x_n^2 = a_2$ cannot be satisfied; this implies that the only vectors that satisfy $\ell_{\text{bin}}(\mathbf{x}) = 0$ from (6) are in the following set:

$$\mathcal{X}_{\text{bin}} = \{ \tilde{\mathbf{x}} \in \{-\alpha, \alpha\}^N : \alpha \in \mathbb{R} \}. \tag{30}$$

The same result would also follow directly from the inspection of (7).

To establish the fact that Regularizer 1 does not have any spurious stationary points (thus, that the regularizer is invex), we need to show that $\nabla \ell_{\text{bin}}(\mathbf{x}) = \mathbf{0}$ iff $\mathbf{x} \in \mathcal{X}_{\text{bin}}$. To this end, we inspect

$$\frac{\partial \ell_{\text{bin}}(\mathbf{x})}{\partial x_n} = 4N x_n^3 - 4[\![\mathbf{x}]\!]^2 x_n = 4x_n (N x_n^2 - [\![\mathbf{x}]\!]^2) = 0, \quad n = 1, \ldots, N. \tag{31}$$

Clearly, every vector $\mathbf{x} \in \mathcal{X}_{\text{bin}}$ satisfies (31). For any other vector, the gradient is nonzero. To prove this, it is sufficient to show that the derivative is nonzero for a two-dimensional, non-binary-valued vector, because any vector with a non-binary-valued subvector is non-binary-valued (and any non-binary-valued vector has a non-binary-valued subvector). To this end, let $\mathbf{x} = [\alpha, \beta]^\mathsf{T}$ for $\alpha \neq \beta$, $\alpha \neq -\beta$, and $\alpha, \beta \neq 0$. Then, we have

$$\frac{\partial \ell_{\text{bin}}(\mathbf{x})}{\partial x_1} = 4\alpha(2\alpha^2 - (\alpha^2 + \beta^2)) = 4\alpha(\alpha^2 - \beta^2) \neq 0, \tag{32}$$

which concludes our proof.

## A.4 DERIVATION OF REGULARIZER 2

Regularizer 2 is minimized by vectors $\mathbf{x} \in \mathbb{R}^N$ that satisfy the linear dependence condition $\mathbf{g}(\mathbf{x}) \sim \mathbf{h}(\mathbf{x})$ for the specific choices $\mathbf{g}(\mathbf{x}) = [x_1^2, \ldots, x_N^2]^{\mathsf{T}}$ and $\mathbf{h}(\mathbf{x}) = \mathbf{x}$. We have

$$\mathbf{g}(\mathbf{x}) \sim \mathbf{h}(\mathbf{x}) \iff \exists (a_1, a_2) \in \mathbb{R}^2 \setminus \{(0,0)\} : a_1 x_n^2 = a_2 x_n, \quad n = 1, \ldots, N. \tag{33}$$

If $x_n = 0$, then the condition $a_1 x_n^2 = a_2 x_n$ is trivially satisfied. If $x_n \neq 0$, then we inspect $a_1 x_n = a_2$. If $a_1 \neq 0$, then $x_n = a_2/a_1$, which implies $x_n = \alpha$ for some $\alpha \in \mathbb{R}$. If $a_1 = 0$ then $a_2 \neq 0$, so the condition $a_1 x_n = a_2$ cannot be satisfied; this implies that the only vectors that satisfy $\ell_{\mathrm{osb}}(\mathbf{x}) = 0$ from (8) are in the following set:

$$\mathcal{X}_{\mathrm{osb}} = \big\{ \tilde{\mathbf{x}} \in \{0, \alpha\}^N : \alpha \in \mathbb{R} \big\}. \tag{34}$$

The same result would also follow directly from the inspection of (9).

To establish the fact that Regularizer 2 does not have any spurious stationary points (thus, that the regularizer is invex), we need to show that $\nabla \ell_{\mathrm{osb}}(\mathbf{x}) = \mathbf{0}$ iff $\mathbf{x} \in \mathcal{X}_{\mathrm{osb}}$. To this end, we inspect

$$\frac{\partial \ell_{\mathrm{osb}}(\mathbf{x})}{\partial x_n} = 2x_n \big( [\![\mathbf{x}]\!]^4 + 2x_n^2 [\![\mathbf{x}]\!]^2 - 3x_n [\![\mathbf{x}]\!]^3 \big) = 0, \quad n = 1, \ldots, N. \tag{35}$$

Clearly, every vector $\mathbf{x} \in \mathcal{X}_{\mathrm{osb}}$ satisfies (35). For any other vector, the gradient is nonzero. To prove this, it is sufficient to show that the derivative is nonzero for a two-dimensional, non-one-sided-binary-valued (non-OSB) vector, because any vector with a non-OSB subvector is non-OSB (and any non-OSB vector has a non-OSB subvector). Assume $\mathbf{x} = [\alpha, \beta]^{\mathsf{T}}$ for $\alpha \neq \beta$ and $\alpha, \beta \neq 0$. Then $\frac{\partial \ell_{\mathrm{osb}}(\mathbf{x})}{\partial x_1} = 2\alpha\beta^2(2\alpha - \beta)(\alpha - \beta)$, and by symmetry, $\frac{\partial \ell_{\mathrm{osb}}(\mathbf{x})}{\partial x_2} = 2\alpha^2\beta(\alpha - 2\beta)(\alpha - \beta)$; this implies that $\frac{\partial \ell_{\mathrm{osb}}(\mathbf{x})}{\partial x_1}$ and $\frac{\partial \ell_{\mathrm{osb}}(\mathbf{x})}{\partial x_2}$ cannot be zero simultaneously.

## A.5 DERIVATION OF REGULARIZER 3

Regularizer 3 is minimized by vectors $\mathbf{x} \in \mathbb{R}^N$ that satisfy the linear dependence condition $\mathbf{g}(\mathbf{x}) \sim \mathbf{h}(\mathbf{x})$ for the specific choices $\mathbf{g}(\mathbf{x}) = [x_1^3, \ldots, x_N^3]^{\mathsf{T}}$ and $\mathbf{h}(\mathbf{x}) = \mathbf{x}$. We have

$$\mathbf{g}(\mathbf{x}) \sim \mathbf{h}(\mathbf{x}) \iff \exists (a_1, a_2) \in \mathbb{R}^2 \setminus \{(0,0)\} : a_1 x_n^3 = a_2 x_n, \quad n = 1, \ldots, N. \tag{36}$$

If $x_n = 0$, then the condition $a_1 x_n^2 = a_2 x_n$ is trivially satisfied. If $x_n \neq 0$, then we inspect $a_1 x_n^2 = a_2$. If $a_1 \neq 0$, then $x_n^2 = a_2/a_1$, which implies $x_n = \pm\alpha$ for some $\alpha \in \mathbb{R}$. If $a_1 = 0$ then $a_2 \neq 0$, so the condition $a_1 x_n^2 = a_2$ cannot be satisfied; this implies that the only vectors that satisfy $\ell_{\mathrm{osb}}(\mathbf{x}) = 0$ from (10) are in the following set:

$$\mathcal{X}_{\mathrm{ter}} = \big\{ \tilde{\mathbf{x}} \in \{-\alpha, 0, \alpha\}^N : \alpha \in \mathbb{R} \big\}. \tag{37}$$

The same result would also follow directly from the inspection of (11).

To establish the fact that Regularizer 3 does not have any spurious stationary points (thus, that the regularizer is invex), we need to show that $\nabla \ell_{\mathrm{ter}}(\mathbf{x}) = \mathbf{0}$ iff $\mathbf{x} \in \mathcal{X}_{\mathrm{ter}}$. To this end, we inspect

$$\frac{\partial \ell_{\mathrm{ter}}(\mathbf{x})}{\partial x_n} = 2x_n \big( [\![\mathbf{x}]\!]^6 + 3[\![\mathbf{x}]\!]^2 x_n^4 - 4[\![\mathbf{x}]\!]^4 x_n^2 \big) = 0, \quad n = 1, \ldots, N. \tag{38}$$

Clearly, every vector $\mathbf{x} \in \mathcal{X}_{\mathrm{ter}}$ satisfies (38). For any other vector, the derivative is nonzero. To prove this, it is sufficient to show that the derivative is nonzero for a two-dimensional, non-ternary-valued vector, because any vector with a non-ternary-valued subvector is non-ternary-valued (and, any non-ternary-valued vector has a non-ternary-valued subvector). Assume $\mathbf{x} = [\alpha, \beta]$ for $\alpha \neq \beta$, $\alpha \neq -\beta$ and $\alpha, \beta \neq 0$. Then $\frac{\partial \ell_{\mathrm{ter}}(\mathbf{x})}{\partial x_1} = 2\alpha\beta^2(3\alpha^2 - \beta^2)(\alpha^2 - \beta^2)$, and, by symmetry, $\frac{\partial \ell_{\mathrm{ter}}(\mathbf{x})}{\partial x_2} = 2\alpha^2\beta(\alpha^2 - 3\beta^2)(\alpha^2 - \beta^2)$; this implies that $\frac{\partial \ell_{\mathrm{ter}}(\mathbf{x})}{\partial x_1}$ and $\frac{\partial \ell_{\mathrm{ter}}(\mathbf{x})}{\partial x_2}$ cannot be zero simultaneously.

## A.6 DERIVATION OF REGULARIZER 4

Regularizer 4 is minimized by vectors $\mathbf{x} \in \mathbb{R}^N$ that satisfy the linear dependence condition $\mathbf{g}(\mathbf{x}) \sim \mathbf{h}(\mathbf{x})$ for the specific choices $\mathbf{g}(\mathbf{x}) = \mathbf{C}\mathbf{x}$ and $\mathbf{h}(\mathbf{x}) = \mathbf{x}$. By definition, we have that $\mathbf{g}(\mathbf{x}) \sim \mathbf{h}(\mathbf{x})$ if and only if $\mathbf{x}$ is an eigenvector of $\mathbf{C}$.

For the sake of completeness, we provide the stationary point analysis of Regularizer 4 below:

$$\nabla_{\mathbf{x}} \ell_{\mathrm{eig}}(\mathbf{x}) = 2(\|\mathbf{x}\|_2^2 \mathbf{C}^T \mathbf{C} + \|\mathbf{C}\mathbf{x}\|_2^2 \mathbf{I}_N - 2(\mathbf{x}^T \mathbf{C}\mathbf{x}))\mathbf{x} = \mathbf{0}_N. \tag{39}$$

Clearly, every eigenvector $\mathbf{x}$ of $\mathbf{C}$ satisfies (39). While it appears to be unlikely that any other $\mathbf{x}$ would satisfy (39), we are unable to provide a formal proof that rules out the existence of other solutions. Therefore, we refrain from claiming that Regularizer 4 has no spurious stationary points.

## A.7 DERIVATION OF REGULARIZER 5

Regularizer 5 is minimized by matrices $\mathbf{X} \in \mathbb{R}^{N \times K}$ that satisfy the linear dependence condition $\mathbf{g}(\mathbf{x}) \sim \mathbf{h}(\mathbf{x})$ for the specific choices $\mathbf{g}(\mathbf{X}) \triangleq \mathrm{vec}(\mathbf{X}^\mathsf{T}\mathbf{X})$ and $\mathbf{h}(\mathbf{x}) \triangleq \mathrm{vec}(\mathbf{I}_K)$.

To establish the fact that Regularizer 5 does not have any spurious stationary points (thus, that the regularizer is invex), we need to show that $\nabla \ell_{\mathrm{om}}(\mathbf{x}) = \mathbf{0}$ iff $\mathbf{x} \in \mathcal{X}_{\mathrm{om}}$. To this end, we inspect

$$\nabla_{\mathbf{X}} \ell_{\mathrm{om}}(\mathbf{X}) = 4N\mathbf{X}\mathbf{X}^\mathsf{T}\mathbf{X} - 4\|\mathbf{X}\|_\mathrm{F}^2\mathbf{X} = \mathbf{0} \tag{40}$$

$$\Rightarrow 4\mathbf{X}(K\mathbf{X}^\mathsf{T}\mathbf{X} - \|\mathbf{X}\|_\mathrm{F}^2\mathbf{I}_K) = \mathbf{0} \tag{41}$$

$$\Rightarrow K\mathbf{X}^\mathsf{T}\mathbf{X} = \|\mathbf{X}\|_\mathrm{F}^2\mathbf{I}_K, \tag{42}$$

which implies that $\mathbf{X}$ must have orthogonal columns of equal length.

## A.8 PROOF OF PROPOSITION 2

From the triangle inequality and Hölder's inequality (Hölder, 1889) follows that

$$|\langle \mathbf{g}(\mathbf{x}), \mathbf{h}(\mathbf{x}) \rangle| \leq \sum_{n=1}^N \left| [\mathbf{g}(\mathbf{x})]_n [\mathbf{h}(\mathbf{x})]_n \right| \leq \|\mathbf{g}(\mathbf{x})\|_p \|\mathbf{h}(\mathbf{x})\|_q, \tag{43}$$

where $p, q \geq 1$ so that $\frac{1}{p} + \frac{1}{q} = 1$. Raising the left-hand and the right-hand sides of (43) to the power of $r > 0$ and rearranging terms leads to

$$0 \leq \|\mathbf{g}(\mathbf{x})\|_p^r \|\mathbf{h}(\mathbf{x})\|_q^r - |\langle \mathbf{g}(\mathbf{x}), \mathbf{h}(\mathbf{x}) \rangle|^r \triangleq \breve{\ell}(\mathbf{x}). \tag{44}$$

Both inequalities in (43) hold iff $\mathbf{g}(\mathbf{x}) \sim \mathbf{h}(\mathbf{x})$, for which $\breve{\ell}(\mathbf{x}) = 0$.

# B GENERALIZATIONS AND VARIATIONS OF CS REGULARIZERS

## B.1 BEYOND VECTOR TERNARIZATION

We now show one approach that generalizes CS regularizers to a symmetric, discrete-valued set with $2^B$ equispaced entries. The idea behind this approach is as follows: (i) decompose $\mathbf{x} \in \mathbb{R}^N$ into a sum of $B$ auxiliary vectors $\mathbf{x} = \sum_{b=1}^B \mathbf{y}_b$ with $\mathbf{y}_b \in \mathbb{R}^N$ and (ii) apply one regularization function to the auxiliary vectors $\mathbf{y}_b$, $b = 1, \ldots, B$.

Define $\mathbf{g}(\{\mathbf{y}_b\}_{b=1}^B) = \left[ \tilde{\mathbf{g}}(\mathbf{y}_1)^\mathsf{T}, \ldots, \tilde{\mathbf{g}}(\mathbf{y}_B)^\mathsf{T} \right]^\mathsf{T}$ using $\tilde{\mathbf{g}}(\mathbf{y}) = [y_1^2, \ldots, y_N^2]^\mathsf{T}$ and $\mathbf{h}(\{\mathbf{y}_b\}_{b=1}^B) = \left[ \tilde{\mathbf{h}}_1(\mathbf{y}_1)^\mathsf{T}, \ldots, \tilde{\mathbf{h}}_B(\mathbf{y}_B)^\mathsf{T} \right]^\mathsf{T}$ using $\tilde{\mathbf{h}}_b(\mathbf{y}) = 4^{b-1}\mathbf{1}_N$. Then, Proposition 1 yields the following CS regularizer that promotes symmetric equispaced-valued vectors; see App. B.1.1 for the derivation.

**Regularizer 6** (Symmetric Equispaced). *Let $\mathbf{y}_b \in \mathbb{R}^N$ for $b = 1, \ldots, B$ and define*

$$\ell_{\mathrm{equ}}(\{\mathbf{y}_b\}_{b=1}^B) \triangleq KN\left( \sum_{b=1}^B \llbracket \mathbf{y}_b \rrbracket^4 \right) - \left( \sum_{b=1}^B 4^{b-1} \llbracket \mathbf{y}_b \rrbracket^2 \right)^2 \tag{45}$$

*with $K \triangleq \sum_{b=1}^B 4^{2(b-1)}$. Then, the nonnegative function (45) is only zero for vectors $\mathbf{y}_b \in \{-2^{b-1}\alpha, 2^{b-1}\alpha\}^N \cup \mathbf{0}_N, b = 1, \ldots, B$, for any $\alpha \in \mathbb{R}$; this implies that the sum of these vectors $\mathbf{x} \triangleq \sum_{b=1}^B \mathbf{y}_b$ is in the set $\mathcal{X}_{\mathrm{equ}} \triangleq \{\pm(2k-1)\alpha\}_{k=1}^{2^{B-1}}$ with $|\mathcal{X}_{\mathrm{equ}}| = 2^B$. Furthermore, $\ell_{\mathrm{equ}}$ does not have any spurious stationary points.*

To gain insight into Regularizer 6, we invoke Lemma 1 and obtain

$$\ell_{\mathrm{equ}}(\{\mathbf{y}_b\}_{b=1}^B) = KN \min_{\beta \in \mathbb{R}} \sum_{b=1}^B \left( ([\mathbf{y}_b]_n - 2^{b-1}\sqrt{\beta})([\mathbf{y}_b]_n + 2^{b-1}\sqrt{\beta}) \right)^2 \tag{46}$$

We also observe this CS regularizer's auto-scale property and only vectors of the form $\mathbf{y}_b \in \{-2^{b-1}\alpha, 2^{b-1}\alpha\}^N$ for some $\alpha \in \mathbb{R}$ minimize (46). This implies that the vectors $\mathbf{x}$ are of the form $\mathbf{x} \in \{-(2^B - 1)\alpha, \ldots, -3\alpha, -\alpha, \alpha, 3\alpha, \ldots, (2^B - 1)\alpha\}^N$ for some $\alpha \in \mathbb{R}$.

In contrast to the initially introduced binarization and ternarization regularizers, Regularizer 6 introduces additional optimization parameters, multiplying the number of the optimization parameters by $B$. We provide an example use case of Regularizer 6 with simulation results for recovering two-bit solutions to underdetermined linear systems in App. C.5.

### B.1.1 DERIVATION OF REGULARIZER 6

Regularizer 6 is minimized by vectors $\{\mathbf{y}_b\}_{b=1}^B$ that satisfy the linear dependence condition $\mathbf{g}(\{\mathbf{y}_b\}_{b=1}^B) \sim \mathbf{h}(\{\mathbf{y}_b\}_{b=1}^B)$ for the specific choices $\mathbf{g}(\{\mathbf{y}_b\}_{b=1}^B) = [\tilde{\mathbf{g}}(\mathbf{y}_1), \ldots, \tilde{\mathbf{g}}(\mathbf{y}_B)]^\mathsf{T}$ using $\tilde{\mathbf{g}}(\mathbf{y}) = [y_1^2, \ldots, y_N^2]^\mathsf{T}$ and $\mathbf{h}(\{\mathbf{y}_b\}_{b=1}^B) = [\tilde{\mathbf{h}}(\mathbf{y}_1), \ldots, \tilde{\mathbf{h}}(\mathbf{y}_B)]^\mathsf{T}$ using $\tilde{\mathbf{h}}(\mathbf{y}_b) = 4^{b-1}\mathbf{1}_N$. We have

$$\mathbf{g}(\{\mathbf{y}_b\}_{b=1}^B) \sim \mathbf{h}(\{\mathbf{y}_b\}_{b=1}^B)$$
$$\Longleftrightarrow \exists (a_1, a_2) \in \mathbb{R}^2 \backslash \{(0,0)\} : a_1 y_{b,n}^2 = a_2 4^{b-1}, \quad b = 1, \ldots, B, \ n = 1, \ldots, N. \quad (47)$$

If $a_1 \neq 0$, then $y_{n,b}^2 = \frac{a_2}{a_1}4^{b-1}$ which implies $y_{n,b} = 0$ or $y_{b,n} = \pm \alpha 2^{b-1}$, $n = 1, \ldots, N$, $b = 1, \ldots, B$, for some $\alpha \in \mathbb{R}$. If $a_1 = 0$ then $a_2 \neq 0$, so the condition $a_1 y_{b,n}^2 = a_2 4^{b-1}$ cannot be satisfied; this implies that the only vectors $\mathbf{y}_b$ that satisfy $\ell_{\mathrm{equ}}(\{\mathbf{y}_b\}_{b=1}^B) = 0$ from (45) are in the following set:

$$\mathcal{Y}_{b,\alpha} = \{-2^{b-1}\alpha, 2^{b-1}\alpha\}^N \cup \mathbf{0}_N, \quad (48)$$

with $\mathbf{y}_b \in \mathcal{Y}_{b,\alpha}, b = 1, \ldots, B$ for any $\alpha \in \mathbb{R}$. The same result would also follow directly from the inspection of (46). Then, the vectors $\mathbf{x} = \sum_{b=1}^B \mathbf{y}_b$ are in the following set:

$$\mathcal{X}_{\mathrm{equ}} = \{\mathbf{x} \in \{-2^{B-1}\alpha, \ldots, -3\alpha, -\alpha, \alpha, 3\alpha, \ldots, 2^{B-1}\alpha\}^N : \alpha \in \mathbb{R}\}. \quad (49)$$

To establish the fact that Regularizer 6 does not have any spurious stationary points (thus, that the regularizer is invex), we need to show that $\nabla \ell_{\mathrm{equ}}(\mathbf{y}_b) = \mathbf{0}, b = 1, \ldots, B$ iff $\mathbf{y}_b \in \mathcal{Y}_{b,\alpha}, b = 1, \ldots, B$ for any $\alpha \in \mathbb{R}$. To this end, we inspect

$$\frac{\partial \ell_{\mathrm{equ}}(\{\mathbf{y}_b\}_{b=1}^B)}{\partial [\mathbf{y}_b]_n} = 4[\mathbf{y}_b]_n \left( KN([\mathbf{y}_b]_n)^2 - 4^{b-1}\sum_{\tilde{b}=1}^B 4^{\tilde{b}-1}[\![\mathbf{y}_{\tilde{b}}]\!]^2 \right) = 0 \quad (50)$$

for $n = 1, \ldots, N$ and $b = 1, \ldots, B$. Clearly, $\mathbf{y}_b \in \mathcal{Y}_{b,\alpha}, b = 1, \ldots, B$ satisfies (50). For a set of vectors in any other form, the derivative is nonzero. To prove this, it is sufficient to show that the derivative is nonzero for a pair of scalars (i.e., $N = 1$) $(y_b, y_{b'})$ for $y_b \neq 0, |y_b| \neq 2^{b-1}|\alpha|$ and $|y_{b'}| = 2^{b'-1}|\alpha|$, because any pair of vectors including these entries would not satisfy (50) (and any set of vectors that do not satisfy (50) must have such a pair of entries). We have that

$$\frac{\partial \ell_{\mathrm{equ}}(y_b, y_{b'})}{\partial y_b} = 4y_b 4^{2(b'-1)}\left(y_b^2 - 4^{b-1}\alpha^2\right) \neq 0, \quad (51)$$

which concludes our proof.

### B.2 BOUNDED CS REGULARIZERS FOR VECTOR BINARIZATION

We now propose alternative binarization regularizers that avoid potential numerical issues caused by higher-order polynomials.

Define $b(x) \triangleq (1+x^2)^{-1}$ and $\mathbf{g}(\mathbf{x}) \triangleq [b(x_1), \ldots, b(x_N)]^\mathsf{T}$. Furthermore, let $\mathbf{h}(\mathbf{x}) \triangleq \mathbf{1}_N$. Then, Proposition 1 yields the following CS regularizer that promotes symmetric binary-valued vectors.

**Regularizer 7** (Bounded Symmetric Binarizer). *Let $\mathbf{x} \in \mathbb{R}^N$ and define*

$$\ell_{\mathrm{bbin}}(\mathbf{x}) \triangleq N \sum_{n=1}^N \frac{1}{(1+x_n^2)^2} - \left(\sum_{n=1}^N \frac{1}{1+x_n^2}\right)^2 \quad (52)$$

*Then, the nonnegative function in (52) is only zero for one-sided binary-valued vectors, i.e., iff $\mathbf{x} \in \{-\alpha, \alpha\}^N$ for any $\alpha \in \mathbb{R}$. Furthermore, $\ell_{\mathrm{bbin}}(\mathbf{x})$ does not have any spurious stationary points.*

An alternative binarization regularizer can be obtained as follows. Define $b(x) \triangleq e^{-x^2}$ and $\mathbf{g}(\mathbf{x}) \triangleq [b(x_1), \ldots, b(x_N)]^\mathsf{T}$. Furthermore, let $\mathbf{h}(\mathbf{x}) \triangleq \mathbf{1}_N$. Then, Proposition 1 yields the following CS regularizer that promotes symmetric binary-valued vectors.

**Regularizer 8** (Alternative Bounded Symmetric Binarizer). *Let $\mathbf{x} \in \mathbb{R}^N$ and define*

$$\ell_{\mathrm{bin,exp}}(\mathbf{x}) \triangleq N \sum_{n=1}^N e^{-2x_n^2} - \left(\sum_{n=1}^N e^{-2x_n^2}\right)^2 \quad (53)$$

*Then, the nonnegative function in (53) is only zero for one-sided binary-valued vectors, i.e., iff $\mathbf{x} \in \{0, \alpha\}^N$ for any $\alpha \in \mathbb{R}$.*

Note that by normalizing (52) and (53) with $1/N^2$, the maximum value of the resulting CS regularizer is bounded from above by 1.

### B.2.1 DERIVATION OF REGULARIZER 7 FROM APP. B.2

Regularizer 7 is minimized by vectors $\mathbf{x} \in \mathbb{R}^N$ that satisfy the linear dependence condition $\mathbf{g}(\mathbf{x}) \sim \mathbf{h}(\mathbf{x})$ for the specific choices $\mathbf{g}(\mathbf{x}) = [(1+x_1)^{-2}, \ldots, (1+x_N)^{-2}]^{\mathsf{T}}$ and $\mathbf{h}(\mathbf{x}) = \mathbf{1}_N$. We have

$$\mathbf{g}(\mathbf{x}) \sim \mathbf{h}(\mathbf{x}) \iff \exists (a_1, a_2) \in \mathbb{R}^2 \backslash \{(0,0)\} : a_1(1+x_n)^2 = a_2, \quad n = 1, \ldots, N. \qquad (54)$$

If $a_1 \neq 0$, then $(1+x_n)^2 = a_2/a_1$ which implies $x_n = \pm\alpha$, $n = 1, \ldots, N$, for some $\alpha \in \mathbb{R}$. If $a_1 = 0$ then $a_2 \neq 0$, so the condition $a_1(1+x_n)^2 = a_2$ cannot be satisfied; this implies that the only vectors that satisfy $\ell_{\text{bbin}}(\mathbf{x}) = 0$ from (52) are in $\mathcal{X}_{\text{bin}}$.

To establish the fact that Regularizer 7 does not have any spurious stationary points (thus, that the regularizer is invex), we need to show that $\nabla \ell_{\text{bbin}}(\mathbf{x}) = \mathbf{0}$ iff $\mathbf{x} \in \mathcal{X}_{\text{bin}}$. To this end, we inspect

$$\frac{\partial \ell_{\text{bbin}}(\mathbf{x})}{\partial x_n} = 4(1+x_n^2)^{-2} x_n \big( -\frac{N}{1+x_n^2} + \sum_{k=1}^{N} \frac{1}{1+x_k^2} \big) = 0, \quad n = 1, \ldots, N. \qquad (55)$$

Clearly, every vector $\mathbf{x} \in \mathcal{X}_{\text{bin}}$ satisfies (55), while the derivative is nonzero for any other vector, similarly to (31).

### B.2.2 DERIVATION OF REGULARIZER 8 FROM APP. B.2

Regularizer 8 is minimized by vectors $\mathbf{x} \in \mathbb{R}^N$ that satisfy the linear dependence condition $\mathbf{g}(\mathbf{x}) \sim \mathbf{h}(\mathbf{x})$ for the specific choices $\mathbf{g}(\mathbf{x}) = [e^{-x_1^2}, \ldots, e^{-x_N^2}]^{\mathsf{T}}$ and $\mathbf{h}(\mathbf{x}) = \mathbf{1}_N$. We have

$$\mathbf{g}(\mathbf{x}) \sim \mathbf{h}(\mathbf{x}) \iff \exists (a_1, a_2) \in \mathbb{R}^2 \backslash \{(0,0)\} : a_1 e^{-x_n^2} = a_2, \quad n = 1, \ldots, N. \qquad (56)$$

If $a_1 \neq 0$, then $e^{-x_n^2} = a_2/a_1$ which implies $x_n = \pm\alpha$, $n = 1, \ldots, N$, for some $\alpha \in \mathbb{R}$. If $a_1 = 0$ then $a_2 \neq 0$, so the condition $a_1 e^{-x_n^2} = a_2$ cannot be satisfied; this implies that the only vectors that satisfy $\ell_{\text{bin,exp}}(\mathbf{x}) = 0$ from (53) are in $\mathcal{X}_{\text{bin}}$.

To establish the fact that Regularizer 7 does not have any spurious stationary points (thus, that the regularizer is invex), we need to show that $\nabla \ell_{\text{bin,exp}}(\mathbf{x}) = \mathbf{0}$ iff $\mathbf{x} \in \mathcal{X}_{\text{bin}}$. To this end, we inspect

$$\frac{\partial \ell_{\text{bin,exp}}(\mathbf{x})}{\partial x_n} = 4\big( -Ne^{-2x_n^2} + e^{-x_n^2} \sum_{k=1}^{N} e^{-x_k^2} \big) x_n = 0, \quad n = 1, \ldots, N. \qquad (57)$$

Clearly, every vector $\mathbf{x} \in \mathcal{X}_{\text{bin}}$ satisfies (57), while the derivative is nonzero for any other vector, similarly to (31).

### B.3 CS REGULARIZERS FOR DISCRETE-VALUED VECTORS WITH FIXED SCALE

If one is, for example, interested in promoting binary-valued vectors with predefined scale, i.e., $\mathbf{x} \in \{-\alpha, \alpha\}^N$ but for a given *fixed* value of $\alpha$, then one can use $\mathbf{g}(\mathbf{x}) \triangleq [x_1^2, \ldots, x_N^2, \alpha^2]^{\mathsf{T}}$ and $\mathbf{h}(\mathbf{x}) \triangleq \mathbf{1}_{N+1}$ in (4). We note, however, that this particular binarization regularizer with $\alpha = 1$ has been utilized before in Tang et al. (2017); similar regularizers can be found in Hung et al. (2015); Darabi et al. (2019). In general, the idea of augmenting the functions $\mathbf{g}$ and $\mathbf{h}$ with constants removes the auto-scale property of CS regularizers.

### B.4 NON-DIFFERENTIABLE CS REGULARIZERS

One can also develop non-differentiable variants of CS regularizers. For example, by defining $\mathbf{g}(\mathbf{x}) \triangleq \big[|x_1|, \ldots, |x_N|\big]^{\mathsf{T}}$ and $\mathbf{h}(\mathbf{x}) \triangleq \mathbf{1}_N$ in Proposition 1, one obtains the CS regularizer

$$\tilde{\ell}_{\text{bin}}(\mathbf{x}) \triangleq N[\![\mathbf{x}]\!]^2 - \|\mathbf{x}\|_1^2, \qquad (58)$$

which also promotes symmetric binary-valued entries. Intriguingly, this regularizer is equal to the scaled empirical variance of the entry-wise absolute values of $\mathbf{x} \in \mathbb{R}^N$, i.e., $\tilde{\ell}_{\text{bin}}(\mathbf{x}) = N^2 \text{Var}(|\mathbf{x}|)$. One could also combine the idea of (58) with Proposition 2 using $p = q = 2$ and $r = 1$ to obtain

$$\check{\ell}_{\text{bin}}(\mathbf{x}) \triangleq \sqrt{N}\|\mathbf{x}\|_2 - \|\mathbf{x}\|_1, \qquad (59)$$

which also promotes symmetric binary-valued entries. Such alternative versions might result in better empirical convergence if, for example, they are used within auto-differentiation frameworks that allow for non-differentiable functions. We conclude by noting that the specific regularizer in (58) has been used in Taner & Studer (2021) for dynamic-range reduction of complex-valued data in wireless systems.

Table 1: Comparison of regularizers for vector binarization. Advantages are designated by (+) and disadvantages by (-).

| Regularizer | Differentiable (+) | Scale-adaptive (+) | Requires additional optimization variables (-) |
|---|---|---|---|
| $\sum_n(|x_n|-1)^2$ | No | No | No |
| $\sum_n(|x_n|-\beta)^2$ | No | Yes | Yes |
| $\sum_n(x_n^2-1)^2$ | Yes | No | No |
| $\sum_n(x_n^2-\beta)^2$ | Yes | Yes | Yes |
| Ours ($\ell_{\text{bin}}$) | Yes | Yes | No |

### B.5 SCALE-INVARIANT HÖLDER REGULARIZER

By slightly modifying the proof of Proposition 2, one can also develop Hölder regularizers that are *scale-invariant*, i.e., in which scaling the entire vector-valued function $\mathbf{g}(\mathbf{x})$ or $\mathbf{h}(\mathbf{x})$ with a nonzero constant has no impact on the regularizer's function value.

**Proposition 3.** *Fix two vector-valued functions* $\mathbf{g}, \mathbf{h} \colon \mathbb{R}^N \to \mathbb{R}^N$ *and define* $\mathcal{X}$ *as in (3). Let* $p, q \geq 1$ *so that* $\frac{1}{p} + \frac{1}{q} = 1$ *and let* $r > 0$. *Furthermore, set* $\varepsilon \geq 0$. *Then, the nonnegative function*

$$\bar{\ell}(\mathbf{x}) \triangleq \frac{\|\mathbf{g}(\mathbf{x})\|_p^r \|\mathbf{h}(\mathbf{x})\|_q^r + \varepsilon}{|\langle \mathbf{g}(\mathbf{x}), \mathbf{h}(\mathbf{x}) \rangle|^r + \varepsilon} - 1 \tag{60}$$

*is zero iff* $\mathbf{x} \in \mathcal{X}$.

The proof of Proposition 3 follows that of App. A.8, but where we first add $\varepsilon \geq 0$ to the left-hand and right-hand sides of (43), followed by a division by $|\langle \mathbf{g}(\mathbf{x}), \mathbf{h}(\mathbf{x}) \rangle| + \varepsilon$ and rearranging terms to arrive at $\bar{\ell}(\mathbf{x})$ in (60).

Such scale-invariant regularization functions require special attention. First, while the parameter $\varepsilon > 0$ prevents the denominator in (60) from becoming zero, only $\varepsilon = 0$ leads to a scale-invariant regularizer. Second, regularizers derived from Proposition 3 may have significantly more spurious stationary points than those obtained via Proposition 2. Third, evaluating the gradient of regularizers derived from (60) is typically more involved. Nonetheless, their (approximately) scale-invariant property might turn out to be useful in some applications and outweigh the above drawbacks. We note that scale-invariant versions of CS regularizers can also be obtained as a special case of Proposition 3.

### B.6 VECTORS IN THE NULLSPACE OF A GIVEN MATRIX

As our last regularizer, we propose a variant that promotes unit-norm vectors in the nullspace of a given (and fixed) matrix $\mathbf{C} \in \mathbb{R}^{M \times N}$. Define $\mathbf{g}(\mathbf{x}) \triangleq [(\mathbf{C}\mathbf{x})^{\text{T}}, \|\mathbf{x}\|_2^2 - 1, 1]^{\text{T}}$ and $\mathbf{h}(\mathbf{x}) \triangleq [\mathbf{0}_{M \times 1}^{\text{T}}, 0, 1]^{\text{T}}$. Then, Proposition 1 yields the following CS regularizer that promotes unit-norm vectors in the nullspace of $\mathbf{C}$.

**Regularizer 9** (Nullspace Vector). *Fix* $\mathbf{C} \in \mathbb{R}^{M \times N}$ *with* $M \geq N$ *and let* $\mathbf{x} \in \mathbb{R}^N$. *Define*

$$\ell_{\text{ns}}(\mathbf{x}) \triangleq \|\mathbf{C}\mathbf{x}\|_2^2 + (\|\mathbf{x}\|_2^2 - 1)^2 \tag{61}$$

*Then, the nonnegative function in (61) is zero for only unit-norm vectors* $\mathbf{x}$ *in the nullspace of* $\mathbf{C}$, *i.e., iff* $\mathbf{C}\mathbf{x} = \mathbf{0}_{M \times 1}$ *with* $\|\mathbf{x}\|_2^2 = 1$.

## C ADDITIONAL RESULTS

### C.1 COMPARISON OF THE BINARIZING CS REGULARIZER WITH EXISTING REGULARIZERS

Table 1 summarizes the key properties of existing regularizers from Section 1.3 and how our regularizer can be superior to those, i.e., by being differentiable, scale-adaptive, and avoiding additional optimization parameters.

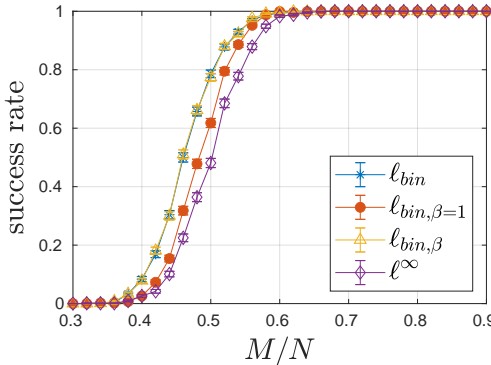

Figure 3: Probability of success in binary solution recovery with the CS regularizer $\ell_{\text{bin}}$ and three baselines.

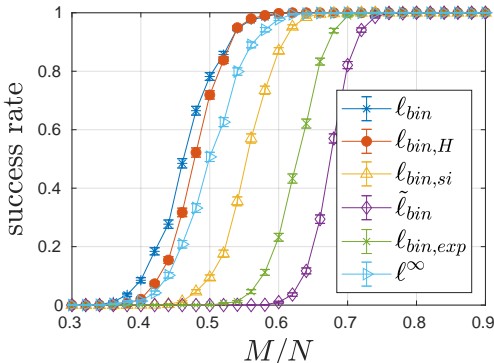

Figure 4: Probability of success in binary solution recovery with five CS regularizer variants and an $\ell^\infty$-norm minimization baseline.

## C.2 Comparisons with Baselines for Binary Recovery

We follow the same experimental setup as in Section 3.1 and provide experiments for two additional baselines: (i) assuming that the scale $\beta$ is known and fixed as a constant and (ii) letting $\beta$ be a separate (and explicit) optimization parameter (that is learned together with the entries of the vector). As mentioned in Section 1.3, we use $\ell_{\text{bin},\beta=1} \triangleq \sum_{n=1}^{N}(x_n^2 - \beta)^2$ with known and fixed $\beta = 1$, and $\ell_{\text{bin},\beta} \triangleq \sum_{n=1}^{N}(x_n^2 - \beta)^2$ with additional optimization parameter $\beta$.

In Fig. 3, we observe that both of these baseline methods achieve comparable recovery performance, but CS regularizers have the advantages of (i) not requiring to know the scale a-priori and (ii) not introducing additional optimization parameters.

## C.3 Additional Results for More CS Regularizers for Binary Recovery

We follow the same experimental setup as Section 3.1 and provide experiments for four additional variants of CS regularizers: Here, $\ell_{\text{bin,H}}$ refers to the Hölder CS regularizer from (17) with $p = q = 2, r = 1$, $\ell_{\text{bin,si}}$ to the scale-invariant Hölder CS regularizer from (60) with $p = q = 2, r = 1$, $\tilde{\ell}_{\text{bin}}$ to the non-differentiable CS regularizer from (58), and $\ell_{\text{bin,exp}}$ to the bounded CS regularizer from (53).

In Fig. 4, we observe that while $\ell_{\text{bin,H}}$ has comparable success rate to $\ell_{\text{bin}}$, the remaining variants are outperformed by the baseline $\ell^\infty$-norm.

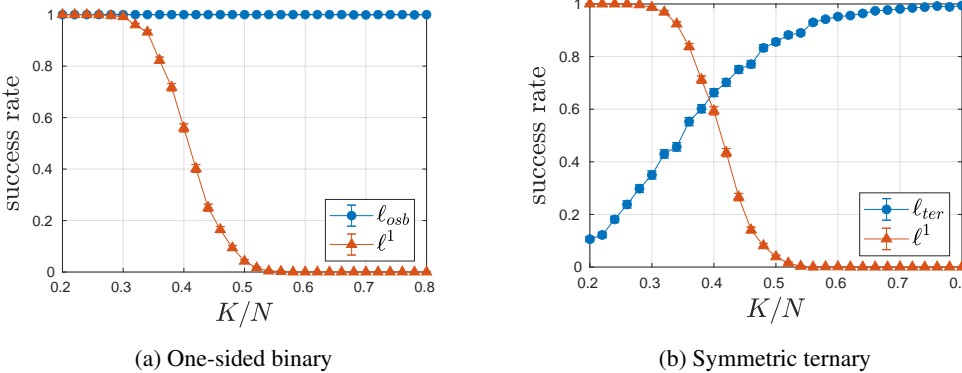

(a) One-sided binary                          (b) Symmetric ternary

Figure 5: Probability of success for recovering vectors with (a) one-sided binary and (b) symmetric ternary values dependent on the density ratio $K/N$.

### C.4 ADDITIONAL RESULTS FOR SPARSE RECOVERY

In this subsection, we slightly modify our experimental setup from Section 3.1 in order to compare the solution recovery performance of $\ell_{\text{osb}}$- and $\ell_{\text{ter}}$-minimization to that of $\ell^1$-norm minimization with respect to the sparsity of $\mathbf{x}^\star$. We fix $N = 100$ and $M = 75$. We create vectors $\mathbf{x}^\star \in \mathbb{R}^N$ with a fixed number of $K$ uniform randomly chosen nonzero entries; these nonzero entries are $+1$ for one-sided binary vectors, and are chosen i.i.d. with uniform probability from $\{-1, +1\}$ for ternary vectors. We vary $K$ from 20 to 80. For each $K$, we randomly generate 1000 problem instances and report the average success probability and the standard deviation from the mean. We only allow one random initialization, and the remaining details of the setup are the same as those presented in Section 3.1.

Fig. 5 demonstrates the success rate of (a) $\ell_{\text{osb}}$-minimization and (b) $\ell_{\text{ter}}$-minimization compared to $\ell^1$-norm minimization with respect to the density ratio of $\mathbf{x}^\star$ given by $\delta = K/N$. In Fig. 5 (a), we observe that while the success rate of $\ell^1$-norm minimization reduces with $\delta > 0.3$ and almost reaches 0 at $\delta = 0.5$, the success rate of $\ell_{\text{osb}}$-minimization is almost always 1 for any density ratio. In Fig. 5 (b), we observe that $\ell^1$-norm minimization follows the same trend as in Fig. 5 (a) as expected, while the success rate of $\ell_{\text{ter}}$-minimization increases with density. The success rate of $\ell_{\text{ter}}$-minimization surpasses that of $\ell^1$-norm minimization for $\delta > 0.4$.

### C.5 SIMULATION RESULTS FOR TWO-BIT SOLUTION RECOVERY FOR REGULARIZER 6

We provide an example use case of the symmetric equispaced regularizer from (45), similarly to those of the binary and ternary regularizers from Section 3.1. To this end, we consider systems of linear equations $\mathbf{b} = \mathbf{Ax}$, where $\mathbf{A} \in \mathbb{R}^{M \times N}$ has i.i.d. standard normal entries and $M < N$. For two-bit-valued solution vector recovery, we create two vectors $\mathbf{y}_1 \in \mathbb{R}^N$ and $\mathbf{y}_2 \in \mathbb{R}^N$ whose entries are chosen i.i.d. with uniform probability from $\{-1, +1\}$ and $\{-2, +2\}$, respectively, and calculate $\mathbf{x}^\star = \mathbf{y}_1 + \mathbf{y}_2$. Then, we calculate $\mathbf{b} = \mathbf{Ax}^\star$, and we try to recover the vector $\mathbf{x}^\star$ from $\mathbf{b}$ by solving optimization problems of the form

$$\hat{\mathbf{y}}_1, \hat{\mathbf{y}}_2 \in \arg \min_{\tilde{\mathbf{y}}_1, \tilde{\mathbf{y}}_2 \in \mathbb{R}^N} \ell_{\text{equ}}(\tilde{\mathbf{y}}_1, \tilde{\mathbf{y}}_2) + \lambda \|\mathbf{A}(\tilde{\mathbf{y}}_1 + \tilde{\mathbf{y}}_2) - \mathbf{b}\|_2^2 \tag{62}$$

using a gradient descent algorithm—specifically, FISTA with backtracking (Beck & Teboulle, 2009; Goldstein) for a maximum of $10^4$ iterations. We fix $N = 10, \lambda = 10^{-5}$ and vary $M$ between 3 and 9. For each $M$, we randomly generate 1000 problem instances, with at most 10 random initializations each, and report the average success probability. We declare success for recovering $\mathbf{x}^\star$ if the returned solution $\hat{\mathbf{x}}$ satisfies $\|\hat{\mathbf{x}} - \mathbf{x}^\star\|_2/\|\mathbf{x}^\star\|_2 \leq 10^{-2}$. Fig. 6 shows the success probabilities with respect to the undersampling ratio $\gamma$ along with (negligibly small) error bars calculated from the standard error of the mean. Here, we observe that $\ell_{\text{equ}}$ can recover symmetric two-bit vectors for large undersampling ratios with a reasonable success probability.

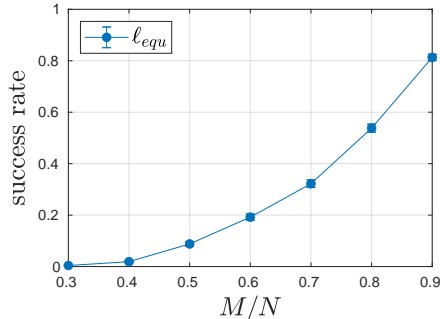

Figure 6: Two-bit solution recovery for $N = 10$ with CS regularizer $\ell_{\text{equ}}$.

## C.6 SIMULATION RESULTS FOR EIGENVECTOR RECOVERY FROM SECTION 3.2

Please see Fig. 7. Here, we observe that minimizing $\ell_{\text{eig}}$ provides a higher success rate than the baseline $\ell_\mu$, which requires an additional optimization parameter compared to $\ell_{\text{eig}}$.

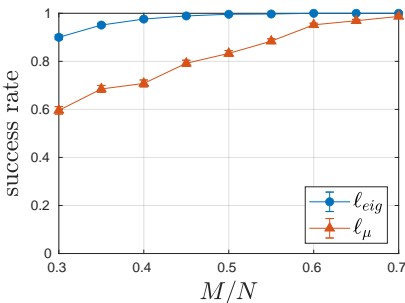

Figure 7: Eigenvector recovery for $N = 100$ with the regularizers $\ell_{\text{eig}}$ and baseline $\ell_\mu$, where $\mu$ must also be learned in the baseline method.

## C.7 APPROXIMATING MAXIMUM-CUT PROBLEMS WITH CS REGULARIZERS

We now showcase another application in which CS regularizers can be utilized. Specifically, CS regularizers can be used to find approximate solutions to the well-known *weighted maximum cut* (MAX-CUT) problem (Commander, 2009). MAX-CUT of a graph is the partition of a graph's vertices into two disjoint sets such that the total weight of the edges between these two sets is maximized. For an undirected weighted graph $G = (V, E)$, this maximization problem can be formulated as the following integer quadratic programming problem:

$$\underset{\mathbf{x} \in \{-1, +1\}^N}{\text{maximize}} \quad \underbrace{\frac{1}{2} \sum_{1 \leq i < j \leq N} w_{ij}(1 - x_i x_j)}_{\triangleq \ell_{\text{MC}}(\mathbf{x})}. \tag{63}$$

Here, $x_i \in \{-1, +1\}$, $i = 1, \ldots, N$, denotes the binary set label for the $i$th vertex of the graph, $N$ is the number of vertices, and $w_{ij} \in \mathbb{R}$ denotes the weight of the edge between the $i$th and $j$th vertices.

The MAX-CUT problem is NP-hard, and many approximations have been proposed in the literature. Classical approximations based on semidefinite and continuous relaxation; see, e.g, Commander (2009) and the references therein. Here, we propose a continuous reformulation that utilizes CS regularizers:

$$\hat{\mathbf{x}} \in \arg \min_{\mathbf{x} \in \mathbb{R}^N} -\ell_{\text{MC}}(\mathbf{x}) + \lambda \ell_{\text{bin}}(\mathbf{x}) \quad \text{subject to } |x_i| \leq 1, \tag{64}$$

which we attempt to solve using a projected gradient descent, similarly to Section 2.2 with a fixed maximum number of iterations.

To evaluate this approximate approach, we ran our projected gradient descent algorithm with random initializations for 10 trials. First, we considered small graphs; here, we set $\lambda = 1$.

- For the $N = 5, E = 7$ graph from Matsuda (2019), we recovered the MAX-CUT solution in less than 300 iterations in each of the 10 trials with random initializations.

- For a graph with $N = 4$ vertices, $E = 5$ edges, and weights $w_{12} = 10$, $w_{13} = 20$, $w_{14} = 30$, $w_{24} = 40$, and $w_{34} = 50$, we recovered the MAX-CUT in less than 40 iterations in each of the 10 trials.

Second, we considered the larger graphs, where $w_{i,j} \in \{-1, 1\}$, given in Matsuda (2019) for benchmarking; here, we set $\lambda = 10^{-7}$. The table below demonstrates the graph ID, the average cut values $\ell_{MC}$ for the initializations and our recovered solutions across 10 trials, and the maximum cut values from Matsuda (2019).

Table 2: Comparison of Our Cut and Known Maximum Cut (Matsuda, 2019) for Different Graphs

| Graph ID | Initial Cut | Our Cut | Known Maximum Cut (Matsuda, 2019) |
|:---:|:---:|:---:|:---:|
| G10 | 67.4±50.5 | 1769.1±26.8 | 2000 |
| G11 | 7.2±14.9 | 482.0±10.46 | 564 |
| G12 | -0.2±24.3 | 470.8±13.7 | 556 |
| G13 | 22.6±25.1 | 494.2±8.1 | 582 |

Our approach struggles to recover the MAX-CUT for large graphs; nonetheless, it significantly improves the objective value compared to the initialization, demonstrating its potential. Moreover, with notable computational advantages over the method in Matsuda (2019), our approach shows promise and could inspire future research.

# D DETAILS OF NEURAL NETWORK QUANTIZATION EXPERIMENTS FROM SECTION 3.4

## D.1 DATASETS AND PREPROCESSING

ImageNet has over 1.2 M training images and 50 k validation images from 1000 object classes. We train and evaluate our network on the training and validation splits, respectively, and report the top-1 accuracy for performance evaluation. We adopt a typical data augmentation strategy on the training images as resizing the shorter side of the images to 256 pixels, taking a random crop of size 224×224, and applying a random horizontal flip. For validation, we apply the same resizing and take the center 224×224 crop.

CIFAR-10 (Krizhevsky, 2009) consists of over 50 k training images and 10 k testing images from 10 object classes. We adopt a typical data-augmentation strategy on the training images as padding by 4 pixels, taking a random 32×32 crop, and applying a random horizontal flip. For testing, we use the original images.

## D.2 THE IMPACT OF VARYING THE REGULARIZATION PARAMETER ON THE CLASSIFICATION ACCURACIES

In Tables 3-6, we provide the average accuracy and standard deviation across 10 runs for various values of $\lambda$. The chosen values are emphasized in bold. Here, we observe that changing the value of $\lambda$ by a factor of 10 does not have a substantial impact on the accuracies.

Table 3: Top-1 accuracy of binarized ResNet-18 on ImageNet for regularized training for various values of $\lambda$.

| $\lambda$ | 0.1 | 1 | 10 | 100 | 1000 |
|---|---|---|---|---|---|
| Top-1 % | 51.5$\pm$0.12 | 59.6$\pm$0.11 | **62.8$\pm$0.09** | 62.4$\pm$0.10 | 59.5$\pm$0.08 |

Table 4: Top-1 accuracy of ternarized ResNet-18 on ImageNet for regularized training for various values of $\lambda$.

| $\lambda$ | 1e3 | 1e4 | 1e5 | 1e6 | 1e7 |
|---|---|---|---|---|---|
| Top-1 % | 63.5$\pm$0.12 | 64.8$\pm$0.10 | **65.3$\pm$0.10** | 65.3$\pm$0.08 | 64.6$\pm$0.17 |

### D.3 PERFORMANCE COMPARISON WITH SOTA BINARIZED AND TERNARIZED NEURAL NETWORKS FROM SECTION 3.4.2

In Table 7, we provide a comparison of the advantages/disadvantages of our training strategy compared to the SOTA methods. We remark that our approach is the only method that does not introduce any additional variables.

In Tables 8-10, we provide a comparison of the image classification accuracy of our methods compared to the SOTA methods. Here, we report the average accuracy and standard deviation for 10 random initializations of training. We remark that the accuracy of our methods is comparable to that of the SOTA methods.

## E COMPUTATIONAL RESOURCES

For our underdetermined linear systems experiments in Section 3.1, we used MATLAB. For the maximum number of $10^4$ iterations, projected gradient descent and Douglas-Rachford splitting algorithms each took approximately one second at most.

For our neural network weight quantization experiments in Section 3.4, we use PyTorch (Paszke et al., 2019). We note that using CS regularizers does not incur a significant additional cost in training; to this end, we measure the time it takes to train for one epoch in the following three scenarios on one NVIDIA RTX 4090 GPU: (i) training with full-precision weights without a regularizer, (ii) regularized training with full-precision weights (Step 1 in Section 3.2.1), and (iii) training with quantized weights (Step 3 in Section 3.2.1). For these three scenarios with ResNet-18, we measured 660 s, 680 s, and 650 s, respectively; with ResNet-20, we measured 4.1 s, 5.2 s, and 3.8 s. These numbers demonstrate that, while the calculation of the CS regularizers naturally results in some overhead in Step 1, the training might be even faster than full-precision training in Step 3, depending on the size of the network and the training dataset. Therefore, depending on the ratio between the number of epochs in Steps 1 and 3, CS-regularizer-based training could be quicker than full-precision training (and thus, all the other SOTA methods) for the same total number of epochs.

## F REGULARIZATION FUNCTION LANDSCAPES

Fig. 8 illustrates the loss landscapes of Regularizer 1, Regularizer 2, and Regularizer 3 in $\mathbb{R}^2$. Here, we clearly see the global minima of each function.

Table 5: Top-1 accuracy of binarized ResNet-20 on CIFAR10 for regularized training for various values of $\lambda$.

| $\lambda$ | 0.1 | 1 | 10 | 100 | 1000 |
|---|---|---|---|---|---|
| Top-1 % | 87.1$\pm$0.17 | 89.5$\pm$0.20 | **90.3$\pm$0.23** | 89.7$\pm$0.18 | 88.4$\pm$0.24 |

Table 6: Top-1 accuracy of ternarized ResNet-20 on CIFAR10 for regularized training for various values of $\lambda$.

| $\lambda$ | 1e3 | 1e4 | 1e5 | 1e6 | 1e7 |
|---|---|---|---|---|---|
| Top-1 % | 90.5$\pm$0.19 | 90.7$\pm$0.17 | **91.0$\pm$0.13** | 91.0$\pm$0.10 | 90.8$\pm$0.17 |

Table 7: Comparison of variables that are required by SOTA neural network quantization methods and CS regularizers (ours) for training. Each column represents variables that are required *in addition* to (unquantized) full-precision neural network training.

| Method | Trainable variables | Non-trainable variables | Tunable hyper-parameters |
|---|---|---|---|
| SQ-BWN (Dong et al., 2017) | Yes | Yes | 0 |
| BWN (Rastegari et al., 2016) | No | Yes | 0 |
| HWGQ (Cai et al., 2017) | No | Yes | 0 |
| PCCN (Gu et al., 2019) | Yes | Yes | 0 |
| BWHN (Hu et al., 2018) | No | Yes | 0 |
| ADMM (Leng et al., 2018) | Yes | No | 1 |
| IR-Net (Qin et al., 2020) | No | Yes | 0 |
| LCR-BNN (Shang et al., 2022) | No | Yes | 2 |
| DAQ (Kim et al., 2021) | No | Yes | 1 |
| ProxyBNN (He et al., 2020) | Yes | Yes | 1 |
| TWN (Li et al., 2016) | No | Yes | 1 |
| QNet (Yang et al., 2019) | No | Yes | 1 |
| QIL (Jung et al., 2019) | Yes | Yes | 0 |
| DoReFa-Net (Zhou et al., 2016) | No | Yes | 0 |
| LQ (Zhang et al., 2018) | Yes | Yes | 0 |
| DSQ (Gong et al., 2019) | Yes | Yes | 0 |
| Ours ($\ell_{\mathrm{bin}}$ and $\ell_{\mathrm{ter}}$) | No | No | 1 |

| Method | Top-1 (%) |
|---|---|
| ResNet-18 (FP) | 69.8 |
| SQ-BWN (Dong et al., 2017) | 58.4 |
| BWN (Rastegari et al., 2016) | 60.8 |
| HWGQ (Cai et al., 2017) | 61.3 |
| PCCN (Gu et al., 2019) | 63.5 |
| BWHN (Hu et al., 2018) | 64.3 |
| ADMM (Leng et al., 2018) | 64.8 |
| IR-Net (Qin et al., 2020) | 66.5 |
| LCR-BNN (Shang et al., 2022) | 66.9 |
| DAQ (Kim et al., 2021) | 67.2 |
| ProxyBNN (He et al., 2020) | 67.7 |
| Ours ($\ell_{\mathrm{bin}}$) | 62.8$\pm$0.09 |

Table 8: Top-1 accuracy of ResNet-18 with binary-valued weights on ImageNet. FP stands for the full-precision model accuracy.

| Method | Top-1 (%) |
|---|---|
| ResNet-18 (FP) | 69.8 |
| TWN (Li et al., 2016) | 61.8 |
| SQ-TWN (Dong et al., 2017) | 63.8 |
| QNet (Yang et al., 2019) | 66.5 |
| ADMM (Leng et al., 2018) | 67.0 |
| LQ (Zhang et al., 2018) | 68.0 |
| QIL (Jung et al., 2019) | 68.1 |
| Ours ($\ell_{\text{ter}}$) | 65.3±0.08 |

Table 9: Top-1 accuracy of ResNet-18 with ternary-valued weights on ImageNet. FP stands for the full-precision model accuracy.

| Method | Top-1 (%) |
|---|---|
| ResNet-20 (FP) | 91.7 |
| DoReFa-Net (Zhou et al., 2016) | 90.0 |
| LQ (Zhang et al., 2018) | 90.1 |
| DSQ (Gong et al., 2019) | 90.2 |
| IR-Net (Qin et al., 2020) | 90.8 |
| DAQ (Kim et al., 2021) | 91.2 |
| LCR-BNN (Shang et al., 2022) | 91.2 |
| Ours ($\ell_{\text{bin}}$) | 90.3±0.17 |
| Ours ($\ell_{\text{ter}}$)[†] | 91.0±0.11 |

Table 10: Top-1 accuracy of ResNet-20 on CIFAR-10 (c) with binary- and ternary-valued weights. FP stands for the full-precision model accuracy.

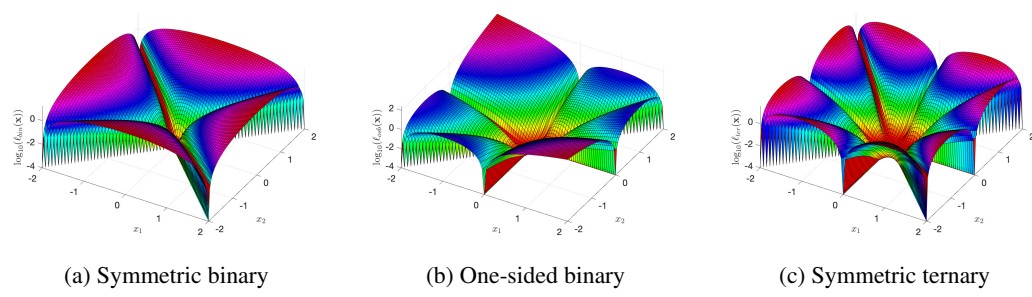

(a) Symmetric binary       (b) One-sided binary       (c) Symmetric ternary

Figure 8: Two-dimensional loss landscape of CS regularizers in logarithmic scale.

