# OpenReview forum: "Cauchy-Schwarz Regularizers"
_ICLR.cc/2025/Conference — ICLR 2025 Poster_

### Official Review · Reviewer_LeuY · 2024-10-23

**Soundness:** 3
**Presentation:** 3
**Contribution:** 3
**Rating:** 8
**Confidence:** 3

**Summary:**

The author(s) study a class of regularization functions that aims to discretize vectors, but also in other cases to find solutions to constrained problems. The derivation is flexible and returns differentiable regularizers (apart from some motivated examples in the appendix). From the idea, some theoretical and practical examples are discussed. This is done especially for cases in which finding discretized versions of solution vectors is of interest, e.g. Neural Network quantization.

**Strengths:**

The paper is well written. Derivation of the regularizers is from first principles. The discussion brought forward after presenting the method has the right balance between practical examples and theoretical examples. It is a nice conference paper idea and looks like there is room for theoretical extensions. Experiments and plots are well exposed. Also, the limitations discussion is very honest.

**Weaknesses:**

- the computational cost might be very high;
- from the paper, one big question/weakness I have (but I will place it here), is: what is the class of constraints that this method can give? This is not discussed in the text, as only explicit examples are shown. We can see it discretizes vectors, we can see it finds solutions to Procrustres-type problems, and we can see it finds eigenvalues. What else?

**Questions:**

- line 097-098: Why would an alternative solution to the orthogonal Procrustres problem be needed? I do not understand the justification.
- figure 4: why is $\ell_{\infty}$ beating some methods?
- why do we observe the "non-convexity of regularizers"-problem only in the bad performance of $\ell_{ter}$ in figure 5? I mean, why not everywhere else? I understand that the ternary regularization is more unstable, but are there other evident reasons?

---

> ### Author Response · Authors · 2024-11-29
>
> We thank the reviewer for their valuable comments and positive feedback!
>
> W1) We now provide the following quantities in order to demonstrate that using CS regularizers is not necessarily incur a significant additional cost: (i) training with full-precision weights without a regularizer, (ii) regularized training with full-precision weights (Step 1 in Section 3.2.1),
> and (iii) training with quantized weights (Step 3 in Section 3.2.1) on one NVIDIA GeForce RTX 4090 GPU. For these three options with ResNet-18, we measured 660 s, 680 s, 650 s, respectively; with ResNet-20, we measured 4.1 s, 5.2 s, 3.8 s.  These numbers demonstrate that, while the calculation of the CS regularizers naturally results in some overhead in Step 1, the training might be even faster than full-precision training in Step 3, depending on the size of the network and the training dataset. Therefore, depending on the ratio between the number of epochs in Steps 1 and 3, CS-regularizer-based training could be quicker than full-precision training (and thus, all the other SOTA methods) for the same total number of epochs. We will include these numbers and observations in the final version of our paper.
>
> W2) To be honest, we do not yet know the general class of constraints (or properties) that our method can give. Nonetheless, our current results already showcase a wide variety of different classes/properties as you have also mentioned. Moreover, based on Reviewer X82’s comment, we now have included yet another application for approximating MAX-CUT problems; please see our response to W1 of Reviewer X82 for more details.
>
> Q1) The well-known solution of the Procrustes problem typically requires computing an SVD, which requires cubic complexity (e.g., with the Golub-Kahan algorithm), while our approach does not. Therefore, our approach can be computationally advantageous, especially for large problems. We will emphasize the complexity advantage of our approach in the revised version of our manuscript.
>
> Q2) Since $\ell_{\infty}$-norm is convex, the convergence of the solution algorithm is guaranteed. As theoretically and empirically shown in Mangasarian & Recht (2011), $\ell^\infty$-norm minimization recovers the binary-valued solution as long as $\gamma=M/N$ satisfies $\gamma>0.5$ and $N$ approaches infinity. We will clarify this in the final version of our manuscript.
>
> Q3) Figure 5b, we observe that our $\ell_{\textnormal{ter}}$ regularizer indeed recovers the correct solution vector as long as the vector is not too sparse, as opposed to the $\ell_1$-norm regularizer. Could the reviewer please clarify what they mean by the “bad performance in Figure 5”?

---

> > ### Comment · Reviewer_LeuY · 2024-11-29
> > **Acknowledgement of Response**
> >
> > Dear author(s),
> > thank you for the response.
> >
> > I have no additional questions.
> > On the side of your request for clarification, I mean exactly that for $\frac{K}{N}\approx 0.4$ your $\ell_{ter}$ minimization surpasses the $\ell_{1}$. On the contrary, we observe a different behavior for $\ell_{osb}$, which is "always better" than $\ell_{1}$. Sorry for not being clear in the first comment, but if I remember correctly my observation was leaning towards the following facts:
> > - this phenomenology is "observed" but not explained by your theory, or maybe it is but I did not notice where, and pointing out where would be nice;
> > - the affirmation "projected gradient descent seems to get stuck at local minima[...]one could perform multiple restarts" might be seen as opaque in terms of validity: meaning that it is a sentence without an experiment to corroborate it.
> >
> > Anyways, it is a minor comment.
> >
> > Thanks!

---

> > > ### Author Response · Authors · 2024-12-03
> > >
> > > Dear Reviewer,
> > >
> > > Thank you for the clarification.
> > >
> > > 1) You are correct that the performance of $\ell_{osb}$ and $\ell_{ter}$ minimization cannot be fully explained by our theory, as a natural consequence of the non-convex objective functions. However, we will mention in the final version our manuscript that the performance of $\ell^1$-norm minimization indeed follows the theoretical expectation, since it is well-known that $\ell^1$-norm minimization helps recover sparse solutions, and the success rate drop for dense solutions is expected. Thank you for pointing out this missing comment.
> > >
> > > 2) Again, thank you for your constructive feedback here! We will include results for multiple restarts in the final version of our manuscript.
> > >
> > > We sincerely appreciate your interest in our manuscript. Thank you very much for helping us improve our final version!

---

### Official Review · Reviewer_ck6X · 2024-11-01

**Soundness:** 4
**Presentation:** 3
**Contribution:** 3
**Rating:** 8
**Confidence:** 4

**Summary:**

The paper introduces a class of regularizers which they coin Cauchy-Schwarz regularizers. This class is fairly general and special cases can be used for promoting binary vectors or eigenvectors. They prove a few properties of these regularizers, e.g. some have no spurious critical points. Numerical experiments then showcase the potential of these regularizers for various applications.

**Strengths:**

The paper is original in its idea and clear its description. It builds upon a broad basis of related research but has its own clear novelty. Both the theoretical and numerical contributions are interesting. Modelling a-priori knowledge into regularizers is not a very timely topic but in my view still important as it gives users more freedom what can easily be incorporated e.g. into training ML models. The discussion is well-written and comprehensive.

**Weaknesses:**

- It is not clear how much impact the proposed work will have. In my view this is a bit niche. This is because, 1) the applications are likely fairly limited and 2) the field has largely moved on from explicit regularizers. That being said, I don't mind that.

- CS regularizers are very related to a branch in imaging inverse problems where structural similar images are encouraged, see e.g. https://royalsocietypublishing.org/doi/full/10.1098/rsta.2020.0205 for an overview.

Key papers in this context are:

Gallardo LA, Meju MA. 2003 Characterization of heterogeneous near-surface materials by joint 2D inversion of DC resistivity and seismic data. Geophys. Res. Lett. 30, 1658. (doi:10.1029/2003GL017370)

Haber E, Holtzman Gazit M. 2013 Model fusion and joint inversion. Surv. Geophys. 34, 675–695. (doi:10.1007/s10712-013-9232-4)

Ehrhardt MJ, Thielemans K, Pizarro L, Atkinson D, Ourselin S, Hutton BF, Arridge SR. 2015 Joint reconstruction of PET-MRI by exploiting structural similarity. Inverse Prob. 31, 015001. (doi:10.1088/0266-5611/31/1/015001)

That being said, both approaches are absolutely complementary, so this does not reduce any novelty but could be mentioned for completeness.

- The first paper gives a fairly slow and repetitive start, e.g. abstract and contributions read very similar and are just a paragraph apart.

**Questions:**

- The property of no spurious critical points is also called invexity. What can be said about minimization of invex functions over convex domains? My guess would be that this is still invex, so the property could be preserved by the constrained formulation. In contrast, if we add a convex data fidelity term, I am certain that this will not lead to an invex problem, hence the usual nonconvex issues arise (as also discussed in the paper)

---

> ### Author Response · Authors · 2024-12-03
>
> We thank the reviewer for their positive feedback and valuable comments!
>
> W1) The reviewer is admittedly correct that our regularizers could be niche, but they are fairly broad. By introducing CS regularizers, we aim to provide a versatile new tool with potential applications across various machine learning and signal processing contexts, hoping to inspire future research.
>
> W2) We sincerely apologize that we could not understand the connection between our approach and the regularizers in the referenced papers. Could the reviewer please elaborate how these might be related? We would be happy to include these references in our final manuscript accordingly.
>
> Q1) While every invex function indeed has no spurious critical points, having no spurious critical points does _not_ imply invexity. We have checked, to start with, whether our $\ell_{bin}$ regularizer is invex according to equation (INV) in [1]; unfortunately, it does not seem to be invex as we could not show that (INV) would hold for all x.
>
> Thank you once again for your interest in our manuscript and your help in improving our final version!
>
> [1] Craven, B.D. (2008). Invexity and its Applications . In: Floudas, C., Pardalos, P. (eds) Encyclopedia of Optimization. Springer, Boston, MA. https://doi.org/10.1007/978-0-387-74759-0_310

---

> > ### Comment · Reviewer_ck6X · 2024-12-03
> >
> > W2) No worries. My apologies for not being specific enough. I was referring to the regularizers defined in eq (4.7). If you choose psi(t) = t^2 and phi(t) = t, then the formulation is very similar to the CS regularizers you propose. In your case, you want to make the vectors g(x) and h(x) "similar". In that work, it is about making the vectors grad f_1(x) and grad f_2(x) similar at every point x in the domain of the functions f_1 and f_2.

---

> > ### Comment · Reviewer_ck6X · 2024-12-03
> >
> > Q1) That is surprising. Can you give me a counterexample (i.e. an example with no spurious critical points which is not invex?) I was always under the impression that the two statements are indeed equivalent, see also Theorem 1 in https://www.cambridge.org/core/journals/anziam-journal/article/what-is-invexity/82C3938EDF3585B2AC5C27093E23814F

---

> > > ### Author Response · Authors · 2024-12-03
> > >
> > > W2) Thank you for the clarification! We now understand that the similarity point of view is for $g(x)$ and $h(x)$, we misunderstood it as if the similarity was about the entries of $x$ (such as binary entries, which all have the same magnitude).
> > >
> > > Q1) We extremely apologize that we missed this reference. You are correct, thank you very much for pointing this out! We will definitely include this in our final manuscript and mention the potential impacts on the minimization problem depending on the nature of the loss function combined with the regularizer.
> > >
> > > Thank you once again for your valuable feedback to improve our manuscript!

---

### Official Review · Reviewer_WyH1 · 2024-11-03

**Soundness:** 3
**Presentation:** 3
**Contribution:** 3
**Rating:** 6
**Confidence:** 4

**Summary:**

This paper introduces a novel class of regularization functions, called Cauchy–Schwarz (CS) regularizers, which can be designed to induce a wide range of properties in solution vectors of optimization problems. The resulting regularizers are (i) simple, (ii) automatically determine the appropriate scale, (iii) free of any spurious critical points, and (iv) differentiable, which enables the use of (stochastic) gradient-based numerical solvers that make them suitable to be used in large-scale optimization problems. Furthermore, the authors also discuss specializations, variations, and generalizations, which lead to an even broader class of new and possibly more powerful regularizers.

**Strengths:**

1. The tool used in this paper is the CS inequality from (Steele, 2004)

2. The authors introduces a novel class of regularization functions, called Cauchy–Schwarz (CS) regularizers.

**Weaknesses:**

1.There are too many refereces in lines 80-84. If you don't compare and explain them one by one, the literature is superfluous.

2.The tool in this paper is the CS inequality from (Steele, 2004). Is there any other advance tools that can be used to desigh such regularization? The authors should compare these methods to show the superiority of CS inequality.

3.The propositon 1 is a known conclusion. A literature is necessary here.

**Questions:**

See weaknesses.

---

> ### Author Response · Authors · 2024-11-29
>
> We thank the reviewer for their valuable comments.
>
> W1) The reviewer correctly points out that lines 80-84 are overly crowded with many references. However, we would like to clarify that each of those references was indeed briefly explained followingly in a cluster of references, based on the similarities among the methods of the references. Therefore, we have decided to slightly improve this paragraph in order to avoid a long list of unexplained references, so that the references only show up where they are explained. Please see our updated manuscript.
>
> W2) Our paper already included the derivation of regularizers with a more general version of the CS inequality, namely the Hölder inequality. Moreover, we have already provided experimental results for such a Hölder regularizer in Appendix C.3. In this specific experiment, the CS regularizer indeed has outperformed the Hölder regularizer. However, we unfortunately cannot provide any theoretical guarantees on which regularizer would be better, as this depends on the specific choice of regularizers and the applications.
>
> W3) We agree that Proposition 1, which serves as our recipe, is an immediate consequence of the CS inequality, as we have mentioned right before the proposition. However, we have not seen the result in Proposition 1 stated in our form in any previous literature despite an extensive literature search. Could you please point us to such literature in case you are aware of any? We would be happy to include any prior art in our final manuscript!

---

### Official Review · Reviewer_X82h · 2024-11-06

**Soundness:** 3
**Presentation:** 3
**Contribution:** 3
**Rating:** 5
**Confidence:** 3

**Summary:**

The paper introduces Cauchy–Schwarz (CS) regularizers. These regularizers can be designed to impose various structural properties in the solution vectors of optimization problems, such as promoting discrete-valued vectors, eigenvectors of a given matrix, or orthogonal matrices. The CS regularizers are differentiable, adapt automatically to appropriate scales, and avoid spurious critical points, making them suitable for gradient-based solvers and large-scale optimization tasks. The authors validate the proposed CS regularizers through applications in under-determined linear systems and neural network weight quantization.

**Strengths:**

The introduction of CS regularizers is the contribution of this paper. The versatility is demonstrated effectively through several concrete examples, including weight quantization and underdetermined linear systems.

Specific CS regularizer examples for various optimization problems are given to make reader easier to follow and use.

**Weaknesses:**

While the applications demonstrated are valuable, they are limited to weight quantization and linear systems. It would have been beneficial to explore other potential applications of CS regularizers in broader machine learning problems.

The paper mentions the nonconvex nature of the proposed regularizers and briefly discusses the presence of spurious critical points for some variants. However, the convergence properties of the optimization algorithm in the presence of these regularizers are not analyzed. More rigorous theoretical or empirical analysis regarding convergence would strengthen the argument for their practical viability.

**Questions:**

I suggest adding experiments to evaluate the performance of CS regularizers in other machine learning contexts beyond neural network quantization,. This could further illustrate the broader applicability of the proposed framework.

It would be helpful to either provide convergence guarantees for specific cases of CS regularizers or at least include empirical evidence demonstrating reliable convergence across multiple trials.

The regularization can be non-convex, e.g., Symmetric Binary case in Sec. 3.1. Using multiple starting points will increase the computational cost and this may be a bottleneck for some large-scale problems.

---

> ### Author Response · Authors · 2024-11-29
>
> We thank the reviewer for their valuable comments.
>
> W1/Q1) Concerning the evaluation of CS regularizers in other machine-learning contexts, we have explored their use to solve maximum-cut (MAX-CUT) problems. We have observed that while using our $\ell_{\textnormal{bin}}$ regularizer can indeed recover optimal graph cuts for small graphs, we have not yet been able to find state-of-the-art MAX-CUT solutions for large graphs. Please see Section C.7 for quantitative results, where we have provided the objective values of some known MAX-CUT (i.e., the target) values, the average of random initializations, and the average of our recovered solutions across 10 trials for certain benchmarking graphs. Although we have not been able to reach the target values for large graphs, we observe that the objective value significantly improves compared to the initialization, demonstrating the practical viability of our approach. Moreover, with possibly notable computational advantages over other MAX-CUT solvers, our approach shows some promise, but more exploration will be necessary.
>
> W2) Concerning the extent of our theory, deriving recovery guarantees (e.g., conditions that guarantee recovery of a given vector when using a CS regularizer) is extremely difficult and will depend on the specific CS regularizer. Nonetheless, we can establish convergence to a critical point with forward-backward splitting (FBS) under certain back-tracking conditions. Concretely, we now include convergence results for FBS applied to optimization problems of the form
> $$\hat{\mathbf{x}} \in \arg\min_{\mathbf{x}\in\mathbb{R}^N} f(\mathbf{x})  + \lambda \ell(\mathbf{x}),$$
> where $f$ is convex (but not necessarily differentiable) and $\ell$ is a (nonconvex) CS regularizer. Note that this setup includes solving underdetermined linear systems of equations as studied in Section 3.1.
>
> For this setup, we can make the following statement:
>
> Theorem:
> Let $q(\mathbf{x})=f(\mathbf{x})+\lambda\ell(\mathbf{x})$ be the objective function of the above optimization problem. Suppose that the step-sizes~$\tau^{(t)}$ of FBS (where $t$ is the iteration index) are bounded away from zero and selected small enough to satisfy the following backtracking line-search condition:
> $$\ell(\mathbf{x}^{(t+1)}) \leq
> \ell(\mathbf{x}^{(t)}) + \langle \mathbf{x}^{(t+1)} - \mathbf{x}^{(t)} ,\nabla \ell(\mathbf{x}^{(t+1))}  \rangle + \frac{1}{2\tau^{(t)}} ||\mathbf{x}^{(t+1)} - \mathbf{x}^{(t)}||_2^2.$$
> Then, the objective $q$ decreases monotonically, i.e., we have
> $$q(\mathbf{x}^{(t+1)}) < q(\mathbf{x}^{(t)}).$$
>
> We are currently working on a refinement of this theorem, which, when further restricting the step-sizes to $\tau^{(t)} < C$ to some constant $C$ that depends on $\lambda$ and the specific CS regularizer, establishes that the sequence of iterates converges. Due to the lack of time, we cannot include the full statement in this response. We expect to include this improved convergence result during the author-reviewer discussion period.
>
> Q2) Concerning theoretical convergence guarantees, we refer the reviewer to our response for W1. Concerning empirical evidence, we would like to emphasize that using CS regularizers with FBS indeed demonstrates reliable convergence depending on the problem setup. For example, please see the high M/N-ratio regimes in Figures 1, 3, and 4, where using CS regularizers recovers the binary solution with probability 1.
>
> Q3) We agree with the reviewer that using multiple restarts could be a bottleneck in large-scale problems. However, we would like to emphasize that our quantized neural network experiments from Section 3.4 are indeed large-scale and we do not require multiple restarts as the standard deviation across different training sessions is quite small in Tables 7-9.

---

### Meta-Review · Area_Chair_A2ay · 2024-12-17

**Metareview:**

The paper introduces Cauchy-Schwarz (CS) regularizers, a versatile class of differentiable regularizers that induce properties like discreteness, eigenvector alignment, and orthogonality. They are spurious-critical-point-free, scalable, and effective for tasks like solving linear systems and neural network weight quantization.

Most of the reviewers believe that the paper makes substantial contributions, and the AC also agrees with this assessment.

**Additional Comments On Reviewer Discussion:**

Four reviewers have evaluated the paper, and their overall assessment is positive. I agree with their evaluation and believe the paper offers a strong contribution with compelling results.

One reviewer noted that, while the demonstrated applications are valuable, they are limited to weight quantization and linear systems, and suggested exploring broader applications of CS regularizers in machine learning. I agree with this suggestion and encourage the authors to summarize their responses in the rebuttal. Additionally, all reviewers raised a few technical questions, which the authors have addressed satisfactorily. I strongly recommend incorporating these remarks into the final version.

---

### Decision · Program_Chairs · 2025-01-22

Accept (Poster)